EMBO
*reports*

# WDR11-mediated Hedgehog signalling defects underlie a new ciliopathy related to Kallmann syndrome

Yeon-Joo Kim[1] (iD), Daniel PS Osborn[1], Ji-Young Lee[1], Masatake Araki[2], Kimi Araki[2], Timothy Mohun[3], Johanna Känsäkoski[4], Nina Brandstack[4], Hyun-Taek Kim[5,†], Francesc Miralles[1] (iD), Cheol-Hee Kim[5], Nigel A Brown[1], Hyung-Goo Kim[6], Juan Pedro Martinez-Barbera[7], Paris Ataliotis[1], Taneli Raivio[4], Lawrence C Layman[6] & Soo-Hyun Kim[1,*] (iD)

## Abstract

WDR11 has been implicated in congenital hypogonadotropic hypogonadism (CHH) and Kallmann syndrome (KS), human developmental genetic disorders defined by delayed puberty and infertility. However, WDR11's role in development is poorly understood. Here, we report that WDR11 modulates the Hedgehog (Hh) signalling pathway and is essential for ciliogenesis. Disruption of WDR11 expression in mouse and zebrafish results in phenotypic characteristics associated with defective Hh signalling, accompanied by dysgenesis of ciliated tissues. *Wdr11*-null mice also exhibit early-onset obesity. We find that WDR11 shuttles from the cilium to the nucleus in response to Hh signalling. WDR11 regulates the proteolytic processing of GLI3 and cooperates with the transcription factor EMX1 in the induction of downstream Hh pathway gene expression and gonadotrophin-releasing hormone production. The CHH/KS-associated human mutations result in loss of function of WDR11. Treatment with the Hh agonist purmorphamine partially rescues the WDR11 haploinsufficiency phenotypes. Our study reveals a novel class of ciliopathy caused by WDR11 mutations and suggests that CHH/KS may be a part of the human ciliopathy spectrum.

**Keywords** ciliopathy; hedgehog signal pathway; hypogonadotropic hypogonadism; kallmann syndrome; WDR11
**Subject Categories** Cell Adhesion, Polarity & Cytoskeleton; Molecular Biology of Disease; Signal Transduction

## Introduction

Motile cilia are essential for fluid transport and confer motility. Non-motile primary cilia are the organizing hub for extracellular signals. Defective ciliogenesis leads to abnormal embryonic development and disrupted tissue homeostasis, resulting in ciliopathies—a group of developmental and degenerative disorders including Bardet–Biedl syndrome (BBS), Joubert syndrome, polycystic kidney disease and Meckel syndrome [1,2]. The severity of ciliopathy phenotypes ranges from generally mild to congenitally lethal and can include hypogonadism, anosmia, obesity, retinal degeneration, renal dysfunction, developmental delay, speech deficit, syndactyly, dental dysgenesis, ataxia, diabetes mellitus and congenital heart disease [3–5]. The cilia are required for the proper delivery of the Hedgehog (Hh) signalling essential for growth, survival, cell fate and embryonic patterning [6]. Hh initiates signalling through activation of Smoothened (SMO) which translocates to primary cilia and modulates the processing of GLI transcription factors [7,8], to induce downstream target genes [9]. Ciliary intraflagellar transport is required for the trafficking of GLI transcription factors to the ciliary tip where SMO is concentrated, and back to the cell body and nucleus [7,8]. Although defective Hh signalling is associated with numerous congenital disorders and cancers [10], its downstream target genes that critically regulate specific organogenesis are poorly understood.

Congenital hypogonadotropic hypogonadism (CHH) and Kallmann syndrome (KS) are classically defined by delayed puberty and infertility secondary to gonadotrophin-releasing hormone (GnRH) dysfunction [11]. However, CHH/KS is a complex

---

1  Molecular and Clinical Sciences Research Institute, St. George's, University of London, London, UK
2  Institute of Resource Development and Analysis, Kumamoto University, Kumamoto, Japan
3  Francis Crick Institute, London, UK
4  Helsinki University Central Hospital, Helsinki, Finland
5  Department of Biology, Chungnam National University, Daejeon, Korea
6  Medical College of Georgia, Augusta University, Augusta, GA, USA
7  Developmental Biology and Cancer Programme, Birth Defects Research Centre, UCL Great Ormond Street Institute of Child Health, London, UK
   *Corresponding author. Tel: +44 2082666198; E-mail: skim@sgul.ac.uk
   †Present address: Max Planck Institute for Heart and Lung Research, Bad Nauheim, Germany

disease, as the patients can also exhibit developmental defects in various organs including the brain, spinal cord, heart, kidney, skeleton, ear and eye, along with intellectual disability, neurosensory and neuromotor anomalies at reduced penetrance [12]. We previously identified WD repeat domain 11 (*WDR11*) as the underlying locus for CHH/KS based on a break point at 10q26.12 and subsequent identification of five missense mutations [13]. A splice site mutation in *WDR11* has also been found in a combined pituitary hormone deficiency disorder [14]. *WDR11* is located within the 600-kb minimal microdeletion region associated with the 10q26 deletion syndrome (MIM 609625) [15,16]. WDR11 contains twelve WD domains—minimally conserved motifs of approximately 40 amino acids forming two beta-propellers potentially serving as molecular scaffolds [13]. We previously identified the homeobox protein EMX1 as a binding partner of WDR11 in a yeast 2-hybrid screen and CHH/KS-associated mutations of WDR11 resulted in failed binding of EMX1 [13]. The nature of their interactions, however, remains unclear. Due to evolutionary conservation and broad tissue expression of WDR11 [13], we hypothesized that it may have roles beyond the development of the reproductive system.

Here we report that WDR11 is involved in the Hh signalling pathway and is essential for normal ciliogenesis. *WDR11*-defective mice and zebrafish exhibit complex developmental abnormalities in multiple organs, resembling features known to be associated with Hh signalling and ciliogenesis, thus linking CHH/KS with ciliopathy spectrum disorders. WDR11 undergoes Hh-dependent intracellular trafficking and interacts with GLI3 and EMX1 to regulate the expression of GnRH. We further show that loss of WDR11 leads to obesity in both mice and men, suggesting that Hh signalling via WDR11 is one of the key links between reproduction and metabolism. *WDR11* may also be an underlying locus for the holoprosencephaly (HPE) spectrum and responsible for many of the phenotypes associated with 10q26 deletion syndrome. Our study proposes a new paradigm for the diagnosis of these genetically overlapping disorders and may expand the disease spectrum of CHH/KS.

## Results

### Generation of *Wdr11* knockout mouse

To better comprehend the biological activities of WDR11, we generated a knockout (KO) mouse line Ayu 21-KBW205 by genetrap mutagenesis. Analysis by 5′-RACE, allele-specific PCR, sequencing and Western blot confirmed the insertion of the pU-21W vector in exon 3 of the mouse *Wdr11* gene, and functional KO of endogenous *Wdr11* expression (Fig 1A, Appendix Fig S1A, Appendix Table S1). There was no disruption of neighbouring genes *Emx2* and *Fgfr2* (Appendix Fig S1B) that are also located within the 10q26 deletion syndrome region and known to be important in forebrain and genitalia development [15,16].

### Expression profiles of *Wdr11* in mouse

*Wdr11* is broadly expressed in various adult organs including the brain, eye, ear, lung, heart, kidney and gonads (Appendix Fig S1). In E10.5–12.5 embryos, *Wdr11* signal is visible in the ventricles of the heart, branchial arches and mesonephric duct. Expression is also detected in the head mesenchyme, developing eye and forebrain (Fig 1B and Appendix Fig S1C). Detailed analysis of the brain revealed *Wdr11* expression in the GnRH neuronal migratory niche including nasal cavity and cribriform plate area in E12.5 embryo as well as the median eminence in the adult brain, showing co-localization with GnRH (Appendix Fig S2 and S3A). Moreover, *Wdr11* is expressed throughout the developing and adult olfactory bulb (OB) with particularly high levels in the glomerular layer (Fig EV1A).

### *Wdr11* deficiency causes retardation of growth and development

Gross morphological examination of the embryos and newborn pups revealed significant developmental defects and growth retardation in the null homozygotes ($Wdr11^{-/-}$), compared to the WT ($Wdr11^{+/+}$) and heterozygote ($Wdr11^{+/-}$) littermates (Fig 1C).

---

Figure 1.  **Generation and characterization of *Wdr11* knockout mouse.**

A  Genotypes were determined by PCR analyses of genomic DNA using specific primers designed for the knockout (KO) or wild-type (WT) alleles. Western blot analyses of total protein lysates extracted from the brain and testis tissue samples demonstrate the absence of endogenous Wdr11 protein at ~130 kDa in the KO. β-actin is a loading control.

B  Sagittal sections of E12.5 $Wdr11^{+/-}$ embryos were stained with antibodies against Wdr11 and x-GAL. Zoomed images are from pituitary (a), branchial arch (b), mesonephric duct (c) and limb bud (d). Scale bars indicate 1 mm in the whole embryo images and 500 μm in the zoomed images.

C  *Wdr11*-deficient mice at various stages exhibit significantly retarded development and growth.

D  Images of $Wdr11^{-/-}$ forelimbs showing abnormal digit separation at E13.5 and digit fusion at P1. Skeletal defects of the mice were demonstrated by double staining with alizarin red (for ossified bones) and alcian blue (for cartilage).

E  H/E-stained coronal sections of E18.5 embryos exhibit multiple defects in forebrain and midline craniofacial structures, and present enlarged ventricles in $Wdr11^{-/-}$. Scale bars, 1 mm. Abbreviations are e, eye; t, tongue; ns, nasal septum.

F  3D reconstruction images of whole embryos at E12.5 by HREM displaying the details of facial features. Scale bar indicates 1 mm.

G  Photographs of 12-week-old brain displaying the OB dysgenesis in the null mutant. Coronal sections of E18.5 $Wdr11^{-/-}$ brain demonstrated a rudimentary OB that failed to separate.

H  3D reconstruction images of HREM sections of $Wdr11^{-/-}$ and WT hearts. Double-outlet right ventricle (two red arrows in upper left panel) is observed in $Wdr11^{-/-}$ compared to the normal outlets of the right and left ventricles (red arrow in bottom panels). Ventricular septal defect (a red arrow in upper right panel) is also observed in $Wdr11^{-/-}$ compared to the normal intact ventricular septum in bottom panel. The septum is marked by double-headed green arrows in all images. Note that the mutant was processed as a whole embryo and the WT as an isolated heart, and thus, poorer dye penetration in the mutant explains the different appearance of the reconstructions.

Source data are available online for this figure.

There was a significant underrepresentation of homozygotes from E13 onwards with less than 6% detected beyond E17.5 (Table 1), suggesting a mid-gestation embryonic lethality. The majority of null mice were either stillborn or died within 1–2 days after birth. Rare null individuals survived through adulthood. The null mutants exhibited abnormal digit separation and syndactyly, as well as shortened limbs and hypoplastic skeletons with reduced or absent bone mineralization. The heads of $Wdr11^{-/-}$ mice showed a diminutive and curved nasal midline and a small lower jaw (Fig 1D).

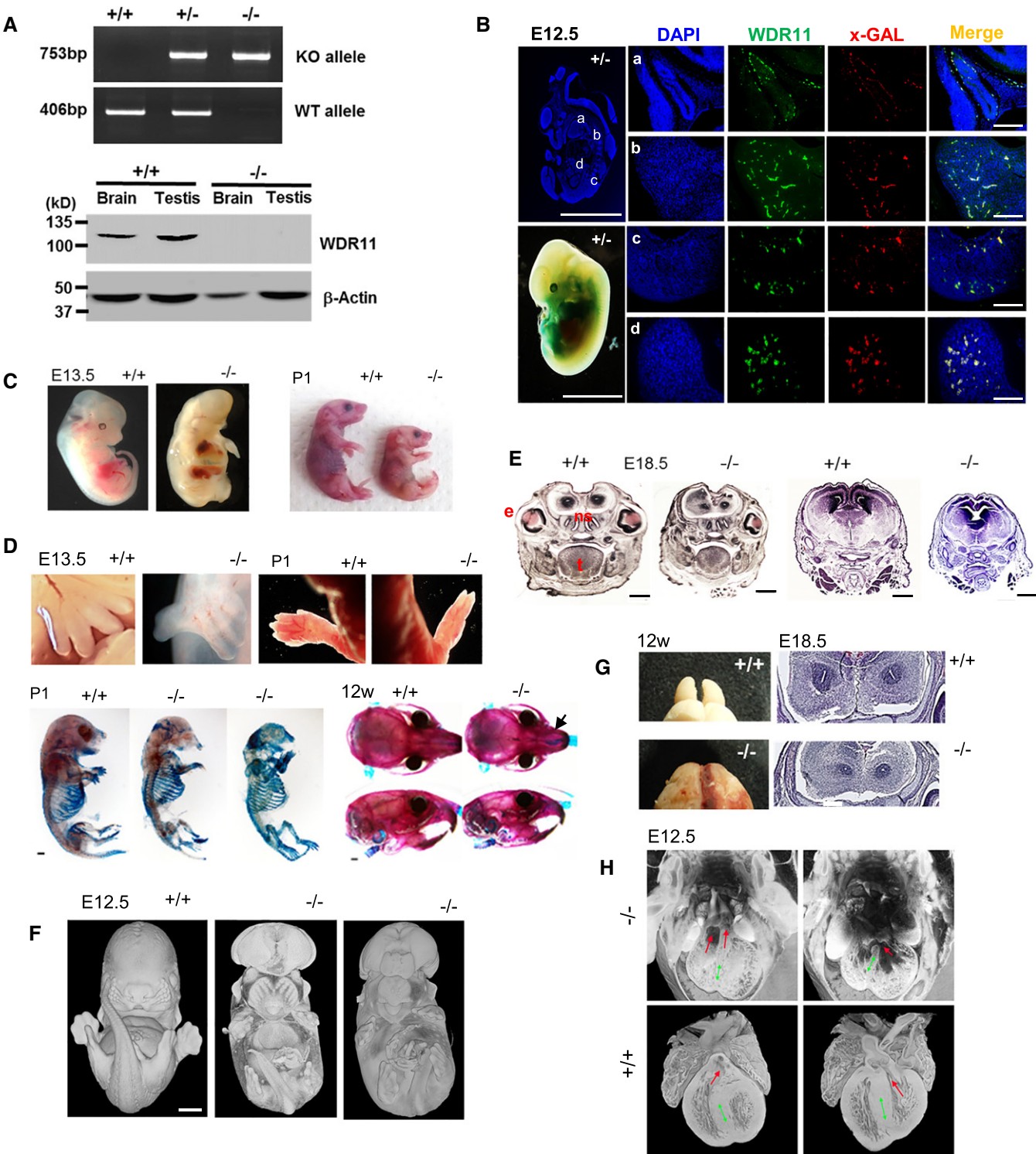

Figure 1.

**Table 1. Mouse phenotype and genotype summary.**

| | Genotype | | |
| --- | --- | --- | --- |
| | *Wdr11*<sup>+/+</sup> | *Wdr11*<sup>+/−</sup> | *Wdr11*<sup>−/−</sup> |
| Expected Mendelian ratio | 25% | 50% | 25% |
| Total observed pups | 34% (86/251) | 61% (153/251) | 5%[a] (12/251) |
| E12.5[b] | 28% (12/43) | 49% (21/43) | 23% (10/43) |
| E14.5 | 30% (18/60) | 65% (39/60) | 5%[c] (3/60) |
| E17.5 | 32% (16/50) | 62% (31/50) | 6%[c] (3/50) |
| Total observed embryos | 30% (46/153) | 59% (91/153) | 10%[c] (16/153) |
| Growth and development delay | | 3% (8/244) | 43% (12/28) |
| Eye defect | | 6% (15/244) | 15% (4/28) |
| Skeletal defect | | | 32% (9/28) |
| Heart defect (embryos) | | | 31% (5/16) |
| Infertility (pups) | | | 75% (9/12) |
| Hydrocephalus (pups) | | | 33% (4/12) |
| Ovary cyst (pups, female) | | | 33% (2/6) |

[a]Significant deviation from the expected Mendelian ratio ($P < 0.01 \times 10^{-9}$) by chi-square test.
[b]No significant deviation from the expected ratio by chi-square test.
[c]Significant deviation from the expected ratio ($P < 0.01$) by chi-square test.

### *Wdr11*-null mice display features of Holoprosencephaly

*Wdr11*-null mutants exhibited several characteristics of holoprosencephaly (HPE) associated with Hh signal deficiency. Most notable was the small head (microcephaly), with closely spaced eyes (hypotelorism) or single/absent eyes (microphthalmia/ anophthalmia), indicative of failed eye field development (Fig 1C and E). In some cases, *Wdr11*-null mutants exhibited exencephaly, indicating a neural tube closure defect, without any sign of spina bifida. The 3D reconstructed images of the mutant embryos by high-resolution episcopic microscopy (HREM) revealed facial features often associated with midline defects such as protruding tongue and fusion of medial nasal processes (Fig 1F). Coronal sections of E18.5 *Wdr11*<sup>−/−</sup> brain further demonstrated midline craniofacial defects, including a partially divided forebrain and narrow mid-facial region, along with the hypoplasia of the neuroepithelium, optic eminence, olfactory pit, nasal septum and Rathke's pouch. The developing *Wdr11*-null brain also displayed enlarged ventricles (Fig 1E). We observed a hypoplastic OB with incomplete separation of the telencephalon similar to that in lobar HPE (Fig 1G).

Congenital heart defects have been reported in CHH/KS [17]. We found a double-outlet right ventricle and ventricular septal defects in E12.5 *Wdr11*<sup>−/−</sup> hearts (Fig 1H). Sometimes thoracic skeletal defects and lung airway abnormalities accompanied an abnormal heart (Movie EV1). The observed frequencies of these phenotypes with respect to the genotypes are shown (Table 1).

### *Wdr11* mutants show hypothalamic GnRH deficiency and pituitary dysgenesis

The total number of GnRH-positive cells was significantly reduced in *Wdr11* KO mice (Fig 2A). Interestingly, in *Wdr11*<sup>−/−</sup> brains, a

**Figure 2. *Wdr11* knockout mouse shows neuroendocrine and metabolic dysfunctions.**

A  Total numbers of GnRH neurons determined by counting the positive immunoreactivity in every fourth section of serially sectioned whole embryo head at E12.5 (WT, $n = 5$; *Wdr11*<sup>−/−</sup>, $n = 5$). Data are presented as means ± SEM. Statistical analysis by unpaired Student's $t$-test (****$P < 0.0001$).

B  Sagittal sections of the brain at E12.5–E14.5 and coronal sections at E18.5 were imaged after H/E staining to investigate pituitary development. rp, Rathke's pouch; vd, ventral diencephalon; m, mesenchyme; inf, infundibulum; rt, rostral tip; al, anterior lobe; il, intermediate lobe; hy, hypothalamus; 3v, third ventricle. Scale bars, 100 μm.

C  Quantitative RT–PCR analyses of genes expressed in the hypothalamus and pituitary of 8- to 10-week-old males ($n = 5$) and dioestrous females ($n = 5$). Data are presented as means ± SEM. Statistical analysis by unpaired Student's $t$-test (ns, $P > 0.05$; *$P < 0.05$; **$P < 0.01$; ***$P < 0.001$; ****$P < 0.0001$). GnRHR, GnRH receptor; LH, luteinizing hormone; FSH, follicle-stimulating hormone; GH, growth hormone; PRL, prolactin.

D  Numbers of pups per litter produced after mating with WT B6 females were tracked in one WT and two null male mice for a period of 7 months. One-way ANOVA with Dunnett's *post hoc* test indicates significantly reduced fertility in the null (***$P < 0.001$; ****$P < 0.0001$). Data presented as means ± SEM.

E  A photograph showing the male testes. H/E-stained sections of testes show hypoplastic Leydig cells and vacuolated seminiferous tubules (asterisk) containing few spermatozoa and spermatids in *Wdr11*<sup>−/−</sup> (Scale bars, 100 μm). Phase-contrast images of the sperm show morphological abnormalities in the null (Scale bars, 20 μm).

F  A photograph showing the female uteri. H/E-stained sections of the uterine wall and ovaries are shown. FL, follicle; CL, corpus luteum; HC, haemorrhagic cyst. Scale bars, 500 μm.

G  Body length (BL) and weight (BW) were measured, and body mass index (BW/BL, g/cm²) was calculated from the measurements at 1 week (WT, $n = 6$; *Wdr11*<sup>+/−</sup>, $n = 6$; *Wdr11*<sup>−/−</sup>, $n = 6$) and 25 weeks (WT, $n = 8$; *Wdr11*<sup>+/−</sup>, $n = 8$; *Wdr11*<sup>−/−</sup>, $n = 6$). Body fat percentages of WT and *Wdr11*<sup>+/−</sup> mice at 25 weeks were calculated by dividing fat weight by body weight. Data are presented as means ± SEM. Statistical analysis by unpaired Student's $t$-test (ns, $P > 0.05$; **$P < 0.01$; ***$P < 0.001$; ****$P < 0.0001$).

Source data are available online for this figure.

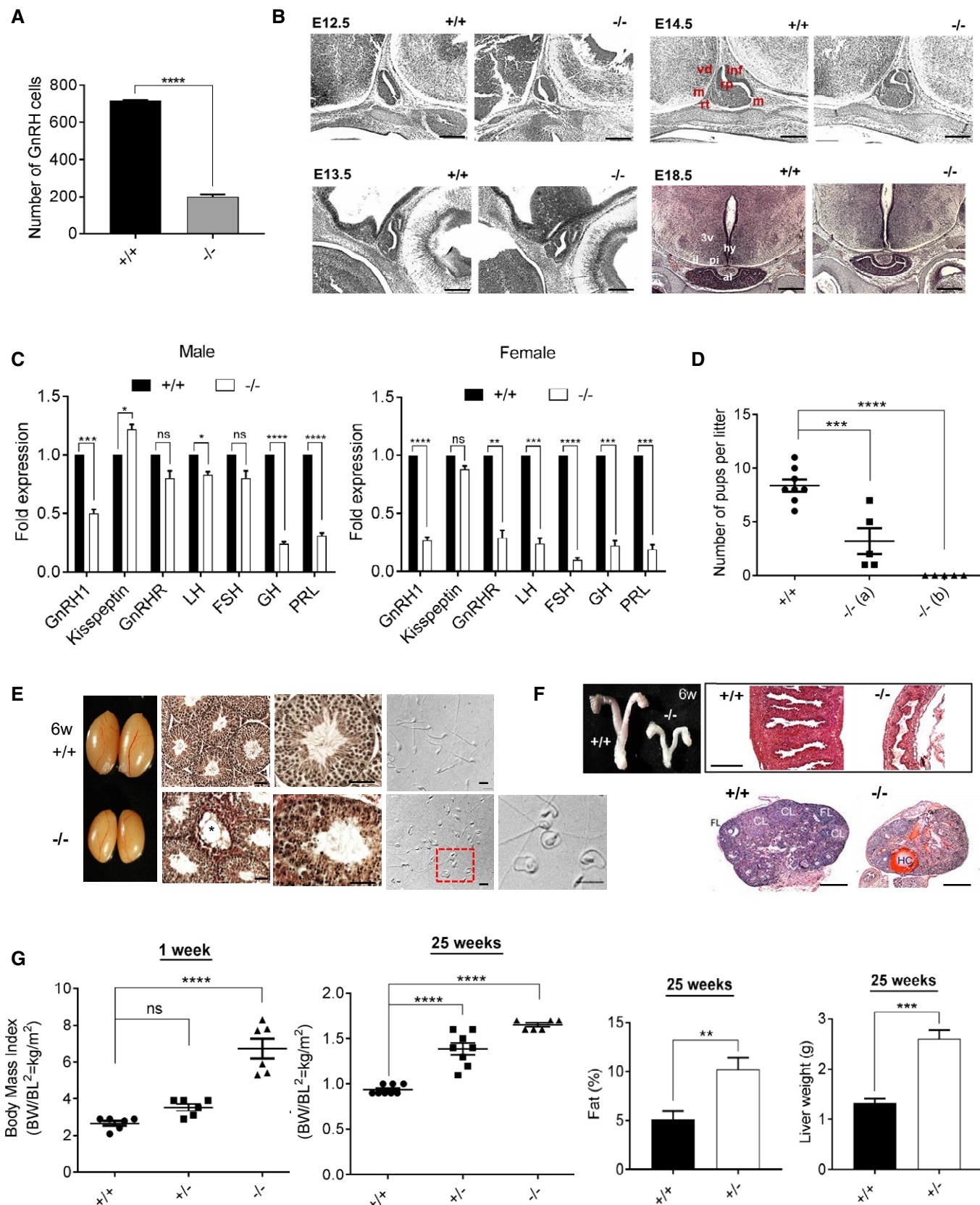

**Figure 2.**

relative accumulation of GnRH-positive cells under the cribriform plate area compared to the forebrain area was observed, potentially suggesting a delayed migration of GnRH neurons (Appendix Fig S3A). *Wdr11*-null embryos also exhibited dysmorphogenesis of the pituitary gland, such as a bifurcation of the anterior lobe and abnormal shaping of the lumen of Rathke's pouch (Fig 2B). Therefore, we analysed the expression of genes essential for normal hypothalamic–pituitary function using samples from 8- to 10-week-old age-matched males and dioestrous females. *Wdr11* KO caused a significant reduction in the expression of *Gnrh1*, *Gnrhr*, *Lhb*, *Fshb*, *Gh* and *Prl* in females. The null males also showed a significant reduction but not in *Gnrhr* and *Fshb* (Fig 2C and Appendix Fig S3B), suggesting that female pituitary functions were more severely affected.

## *Wdr11* deficiency causes delayed puberty, reproductive dysfunctions and obesity

Both male and female null mutants showed hypoplasia of the reproductive organs. *Wdr11*-null male mice exhibited underdeveloped external genitalia and microphallus, along with a reduced anogenital distance index (Appendix Fig S4A). The timing of balanopreputial separation was significantly ($P = 0.00076$; $F_{(2,12)} = 24.68$) delayed in the null ($34 \pm 0.32$ days, $n = 5$), compared to WT ($30 \pm 0.55$ days, $n = 5$) or heterozygotes ($29.4 \pm 0.6$ days, $n = 5$). *Wdr11*-deficient males had smaller testes with abnormal seminiferous tubules containing fewer spermatozoa and spermatids. Notably, *Wdr11* was expressed throughout sperm flagella (Fig EV1B) and loss of *Wdr11* caused a high frequency ($> 50\%$) of morphologically abnormal sperm, leading to subfertility or infertility (Fig 2D and E). In females, the null mutants had smaller ovaries and uteri, with thin and poorly differentiated walls. *Wdr11*$^{-/-}$ ovaries showed arrested follicle development with absent corpora lutea and antral follicles. Widespread atresia and haemorrhagic cysts were observed (Fig 2F). *Wdr11*$^{-/-}$ females exhibited dysregulated oestrous cycles (Appendix Fig S4B).

*Wdr11*-null mice showed substantial growth retardation in terms of body weight (Appendix Fig S4C). However, because they also had a short stature, their body mass index indicated early-onset obesity, while the heterozygotes showed late-onset obesity with signs of a fatty liver (Fig 2G, Appendix Table S3, and Appendix Fig S4D).

## *Wdr11* is localized to the primary cilium and required for ciliogenesis

Since Hh signalling requires functional primary cilia and the phenotypes of *Wdr11*-null mutants resemble ciliopathy spectrum

disorders, we hypothesized that *Wdr11* deficiency disrupts normal ciliogenesis. Mouse embryo fibroblasts (MEFs) derived from *Wdr11*$^{-/-}$ mice exhibited defective ciliogenesis with a significant reduction in the length of the ciliary axoneme and the frequency of ciliated cells, but no defects were found in the formation of the basal body itself (Fig 3A). Since both GnRH neurons and olfactory axons are known to be ciliated *in vivo* [18–20], we examined the effects of *Wdr11* KO on the OB and ME. There was significant deprivation of axoneme structures in these organs in *Wdr11*$^{-/-}$ embryos, indicating disrupted ciliogenesis (Fig EV1A and C). The abnormal choroid plexus present in *Wdr11*$^{-/-}$ brain, which led to hydrocephalus, contained much fewer epithelial cilia compared to the WT, although the overall axonemal structure appeared normal (Fig 3B).

## *Wdr11* knockdown in zebrafish causes developmental defects associated with aberrant Hh signal and ciliogenesis

To further investigate the function of *WDR11* during vertebrate development and ciliogenesis, we employed a zebrafish model system where the endogenous *wdr11* was knocked down using two non-overlapping antisense morpholinos (MOs) targeting the exon 3–intron 3 (E3I3 MO) and exon 9–intron 9 (E9I9 MO) splice sites, independently injected at the 1- to 2-cell stage. E3I3 MO caused an inclusion of intron 3, resulting in a predicted in-frame stop codon in intron 3 (Appendix Fig S5A). E9I9 MO caused an exclusion of exon 9, predicting a premature stop codon in exon 10 (Appendix Fig S5B). When we evaluated the general morphology and survival of the embryos at 72 h post-fertilization (hpf) compared to the control embryos, both MOs resulted in comparable developmental defects known to be associated with cilia dysfunction in zebrafish [21]. These included microphthalmia, microcephaly, melanocyte disorganization, curved body axis, motility defects and narrow trunk (Fig 4A and B). Since Hh is a major regulator of myogenic differentiation in zebrafish [22], we conducted a birefringence assay to investigate myogenesis and myotome/muscle fibre organization. We found a substantial reduction in birefringent signal at 72 hpf in both morphants compared to stage-matched controls, indicating impaired muscle fibre alignment, which could underlie their motility defects (Fig 4C). To determine whether altered Hh signalling contributed to this myogenic phenotype, we assessed the expression level of *ptc1* as readout. At 24 hpf, *wdr11* knockdown (KD) caused a reduced expression of *ptc1* in the cranium and myotome (Fig 4D), indicating that *wdr11* is required for appropriate Hh signalling in these structures. Moreover, *wdr11* morphants exhibited aberrant head cartilage formation and cranial–facial dysmorphology, previously associated

**Figure 3. WDR11 is required for ciliogenesis.**

A  Immunofluorescence images of MEFs derived from E12.5 WT and null embryos. Arl13b stains axoneme and gamma-tubulin (g-TUB) stains the basal bodies of cilia. The percentages of ciliated cells were assessed against the total DAPI-stained cells in random fields. The length of cilia was assessed in 150 cells per genotype by measuring the maximum projection using ImageJ. Data represent mean $\pm$ SEM from three independent experiments using two-tailed unpaired *t*-test (*****$P < 0.00001$). Scale bar, 50 μm.

B  Choroid plexus sections of 8-week-old brain exhibited morphological alterations and lack of ciliary acetylated tubulin (ACT) staining in *Wdr11*$^{-/-}$. Scale bars, 1 mm (main image) and 500 μm (zoomed images). TEM micrographs of longitudinal (1,000×) and horizontal (15,000×) sections of the choroid plexus epithelium show no significant difference in the 9+2 microtubule arrangement of the axoneme.

C  Immunofluorescence analyses of NIH3T3 cells transfected with GFP-tagged WDR11 construct and treated with purmorphamine (Pur), cyclopamine (Cyc), leptomycin B (Lep) and the solvent dimethylformamide (Solv) for 10 h before staining with anti-ACT or anti-g-TUB antibodies followed by DAPI. Scale bar, 10 μm.

D  The percentage of cells showing either nuclear or cytoplasmic localization of WDR11-GFP were quantified in HEK293 cells after each treatment as in (C). Data from four independent experiments, counting 300–400 cells in each experiment, are presented as mean $\pm$ SEM with two-way ANOVA followed by Bonferroni's *post hoc* test (****$P < 0.0001$).

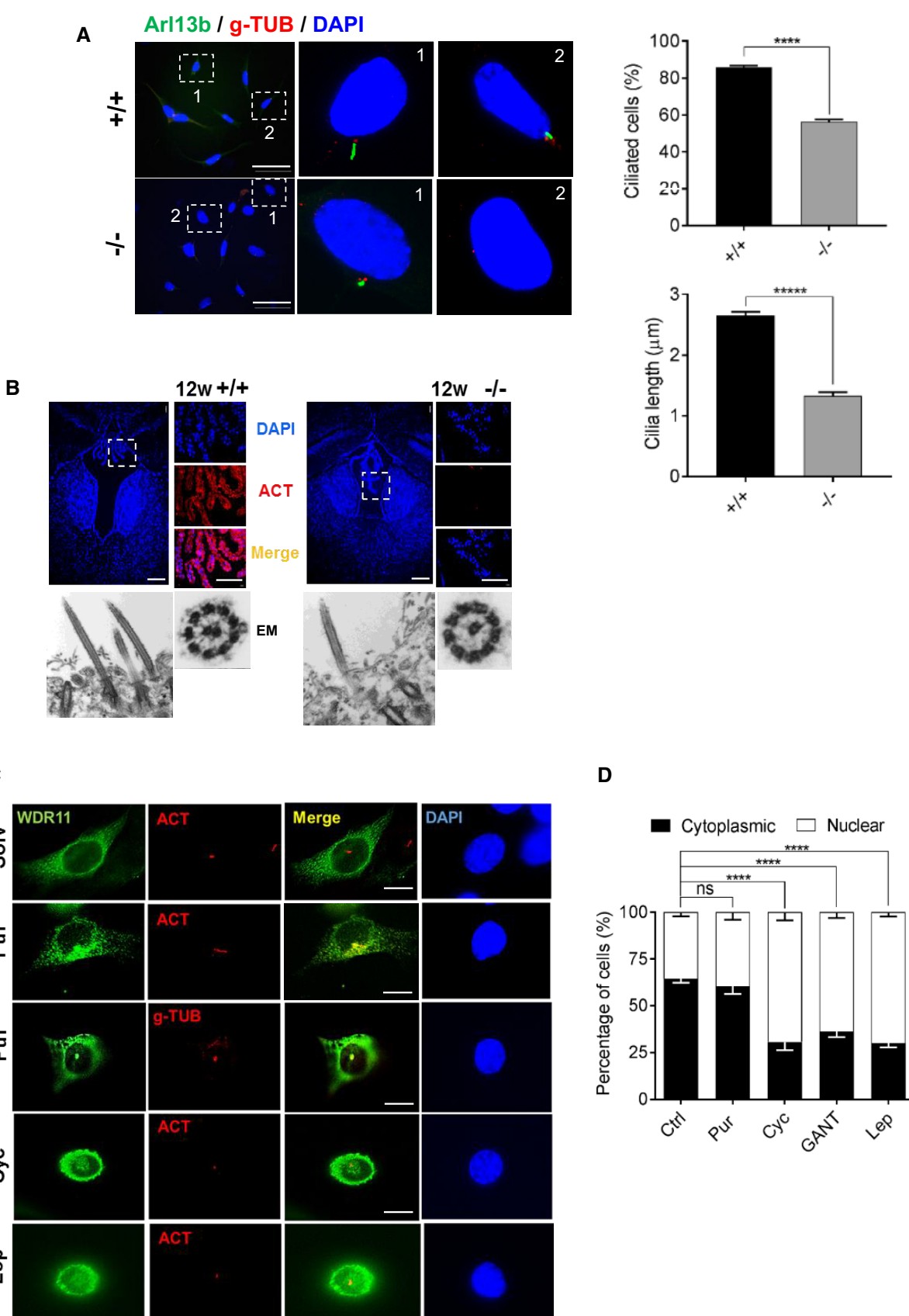

**Figure 3.**

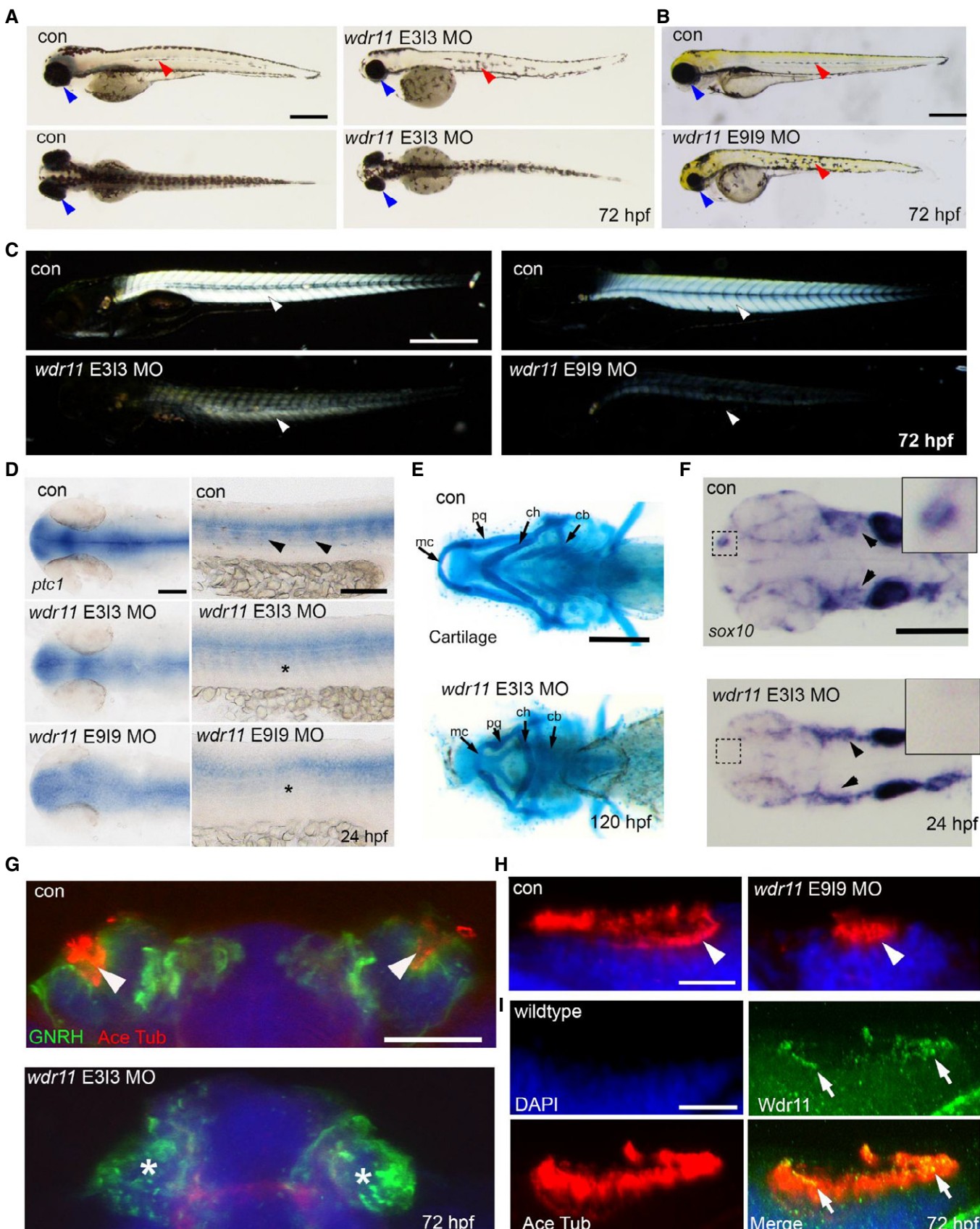

**Figure 4.**

◄

**Figure 4.  Wdr11 knockdown in zebrafish causes defects associated with aberrant ciliogenesis.**

A, B  Antisense morpholinos (MO) targeting either the exon 3–intron 3 (A) or exon 9–intron 9 (B) splice sites of zebrafish *wdr11* result in microphthalmia (blue arrowheads), disorganized melanocyte migration (red arrowheads), a small cranium and reduced myotome formation, compared to control MO or uninjected embryos at 72 h post-fertilization (hpf). Con MO, n = 303/306; *wdr11* E3I3 MO, n = 157/178; *wdr11* E9I9 MO, n = 94/105. Scale bar, 500 μm.

C  Birefringence analysis of *wdr11* morphants compared to control embryos at 72 hpf reveals a requirement for *wdr11* in muscle formation. Arrowheads indicate birefringence signal from skeletal muscle. Con MO, n = 40/40; *wdr11* E3I3 MO, n = 20/20; *wdr11* E9I9 MO, n = 20/20. Scale bar, 500 μm.

D  *Ptc1* expression was reduced in the head (left panels) and myotome (right panel, arrowheads and asterisks) of *wdr11* morphant embryos at 24 hpf, as shown by *in situ* hybridization (ISH). Con MO, n = 84/84; *wdr11* E3I3 MO, n = 95/97; *wdr11* E9I9 MO, n = 60/63. Scale bar, 100 μm.

E  Alcian blue staining of *wdr11* E3I3 morphant embryos, compared to control at 120 hpf. Meckel's cartilage (mc), palatoquadrate (pq), ceratohyal (ch), ceratobranchials (cb) are indicated. Con MO, n = 86/87; *wdr11* E3I3 MO, n = 124/128. Scale bar, 200 μm.

F  *Sox10* ISH indicated a loss and reduction of neural crest cell (NCC) migration to OB (inset) and cranium (arrowheads), respectively, in *wdr11* E3I3 morphants at 24 hpf. Con MO, n = 56/56; *wdr11* E3I3 MO, n = 103/111. Scale bar, 200 μm.

G  Immunofluorescence staining of Gnrh (green) and acetylated tubulin (red) revealed reduced olfactory Gnrh neurons and ablation of cilia (arrowheads) in olfactory pits of E3I3 morphants at 72 hpf. Con MO, n = 20/20; *wdr11* E3I3 MO, n = 18/20. Scale bar, 50 μm.

H  Acetylated tubulin staining (red) marking the cilia of control and E9I9 morphants demonstrates that knockdown of *wdr11* results in reduced ciliogenesis in the olfactory pits (arrowheads). Con MO, n = 20/20; *wdr11* E9I9 MO, n = 19/20. Scale bar, 20 μm.

I  Localization of Wdr11 (green) in the olfactory cilia (red, arrows) of WT embryos at 72 hpf. n = 10. Scale bar, 20 μm.

with Hh and ciliogenesis defects [23,24]. Overall cranial size was short in length with a specific reduction in Meckel's cartilage and palatoquadrate, both components of the lower mandible. Mislocalized development of ceratohyal and ceratobranchial structures was also observed (Fig 4E and Appendix Fig S5C). Since much of the cranial cartilage is neural crest cell (NCC)-derived and melanocyte and head cartilage defects are associated with aberrant NCC migration and defective ciliogenesis [25], we next performed *in situ* hybridization of *sox10* as a marker for the migrating NCC. We found reduced but persistent NCC streaming in the midbrain and pharyngula regions of *wdr11* morphants (Fig 4F and Appendix Fig S5D). However, despite the comparable NCC migration into the diencephalic region, *wdr11* morphants showed a striking absence of *sox10* expression in the OB (Fig 4F, inset). These data suggest that *wdr11* is likely to facilitate cranial NCC migration and is required for olfactory NCC specification.

The zebrafish OB and pronephros are heavily populated by motile cilia, essential for the detection of odorants and fluid flow. E3I3 or E9I9 MO caused a complete loss or reduction of cilia in the olfactory pits, respectively (Fig 4G and H). Despite the absent cilia, GnRH immunoreactivity was still detectable in the OB area of E3I3 morphants, albeit showing an unusually punctate distribution (Fig 4G). Consistent with a role in OB ciliogenesis, *wdr11* was detected in the ciliary axoneme of OB (Fig 4I). Analysis of the pronephric duct revealed defective ciliogenesis in E3I3 morphants, but not in E9I9, suggesting variable, tissue-specific sensitivity between the MOs (Appendix Fig S5F). Notably, basal bodies still remained intact, implying that *wdr11* may be required for cilia extension and growth, but not for nucleation. Loss of *Wdr11* did not affect normal looping of hearts, a laterality defect frequently associated with defects in nodal cilia (Appendix Fig S5E). Combined, these results are in line with our findings in the mouse model and support an evolutionarily conserved role of *wdr11* in Hh signalling and ciliogenesis.

### Intracellular trafficking of WDR11 in response to Hh signal

To further understand the role of WDR11 in ciliogenesis and Hh signalling pathway, we conducted intracellular localization analyses *in vitro*. WDR11 co-stained with markers of the axoneme (acetylated tubulin) or basal bodies/centrosomes (gamma-tubulin) revealed prominent staining of WDR11 around the basal bodies, with

occasional axonemal and nuclear staining. WDR11 also exhibited diffuse cytoplasmic distribution with distinctive perinuclear accumulation in most cells in the basal state (Fig 3C and Appendix Fig S6A). WDR11 contains a putative nuclear export signal (LQNKLPDLSL) raising the possibility that WDR11 shuttles between cytoplasm and nucleus. To determine whether such movement of WDR11 correlates with Hh signalling, we tested the effects of Hh agonist (purmorphamine), Hh antagonists (cyclopamine) and GLI antagonist (GANT61) in NIH3T3 and HEK293 cells. Purmorphamine treatment did not change the localization pattern of WDR11 compared to solvent treatment, but it stimulated cilia growth. Interestingly, both cyclopamine and GANT61 caused a distinct nuclear accumulation of WDR11, which was comparable to nuclear export inhibitor leptomycin B treatment. In contrast, EMX1 remained in the nucleus regardless of treatments (Fig 3D and Appendix Fig S6A), indicating that although EMX1 is a binding partner of WDR11, it does not follow the same intracellular trafficking.

### WDR11 forms a tertiary complex with EMX1 and GLI3 and regulates *GNRH1* and Hh pathway gene expression

Because WDR11 itself does not contain a nuclear localization signal and its binding partner EMX1 remains exclusively nuclear, we speculated that GLI proteins might mediate WDR11's nuclear trafficking upon Hh signalling. Reciprocal co-immunoprecipitation analyses demonstrated that WDR11 could bind to both full-length (FL) and cleaved repressor (R) form of GLI3, in overexpressed forms in HEK293 (Fig 5A) as well as the endogenous forms in mouse tissue lysates (Fig 5B). We also found that EMX1 was a part of this complex, suggesting the formation of a tertiary complex of WDR11/GLI3/EMX1. However, EMX1 alone could not bind to GLI3 in the absence of WDR11, as EMX1/GLI3 binary complex could not be precipitated from *Wdr11*$^{-/-}$ MEFs, contrary to the WT MEFs or HEK293 cells which express endogenous WDR11 (Fig 5C). Therefore, WDR11 is essential for EMX1-GLI3 interaction.

We hypothesized that WDR11 could modulate the transcriptional activity of these two transcription factors. *Gli3* was reported to regulate *Emx1/2* expression in the developing mouse telencephalon [26,27]. Notably, the consensus GLI-binding motif is present in the 5′ upstream region of *EMX1* (GTG*GATCTCCCA*GTG at -692) in the human genome. The *GNRH1* promoter sequences do not contain

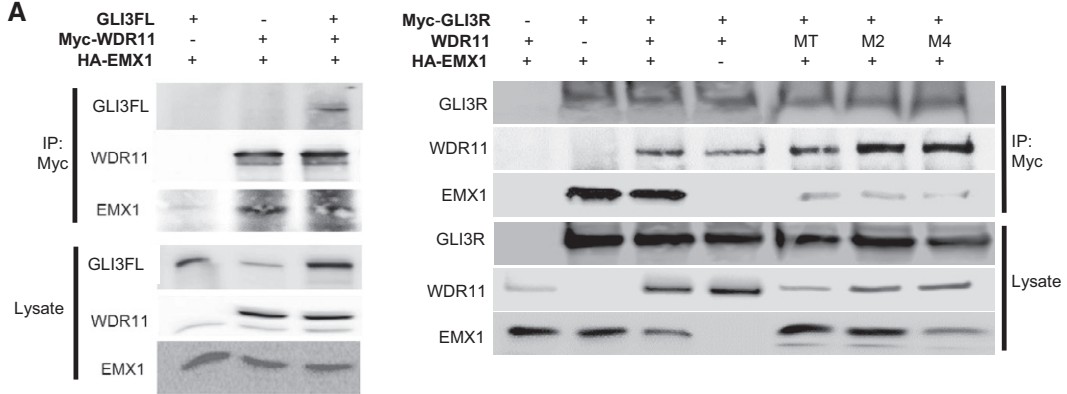

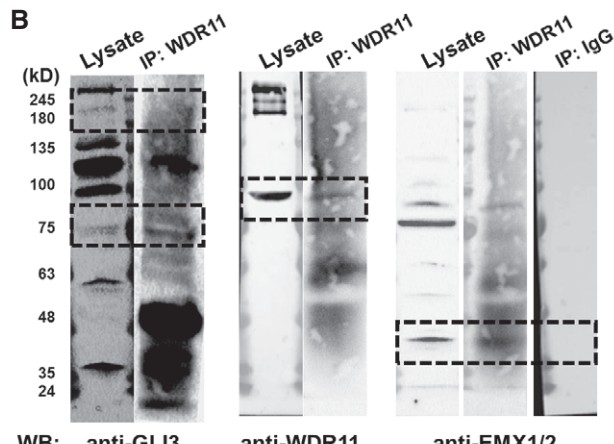

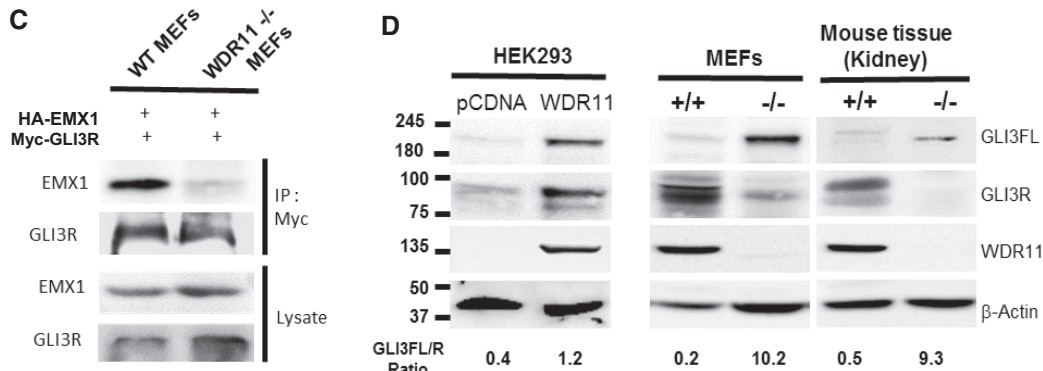

**Figure 5.  WDR11 forms a tertiary complex with EMX1 and GLI3.**

A  Co-immunoprecipitation (co-IP) analyses of HEK293 cells co-transfected with 6xMyc-tagged WDR11 and HA-tagged EMX1 expression constructs either alone or together with GLI3 full-length (GLI3FL) expression construct, demonstrating the formation of WDR11-EMX1-GLI3FL tertiary complex (left panel). In a reciprocal co-IP experiment, 6xMyc-tagged GLI3R was co-transfected with HA-tagged EMX1 either alone or together with untagged WDR11 expression construct to demonstrate the formation of GLI3R-EMX1-WDR11 complex. The disease-associated mutant WDR11 showed a reduced binding capacity to EMX1 within this complex (right panel).

B  Co-IP analyses of mouse testes tissue lysates showed that endogenous WDR11 could bind to endogenous GLI3FL (190 kDa), GLI3R (83 kDa) and EMX1/2 (35 kDa) proteins. Tissue lysates were precipitated with anti-WDR11 antibody or non-immune IgG, and probed with antibodies against GLI3, WDR11 or EMX1/2.

C  6xMyc-GLI3R and HA-EMX1 were co-transfected in WT or *Wdr11*⁻/⁻ MEFs and precipitated with anti-Myc antibody and probed with EMX1 and GLI3 antibodies. EMX1-GLI3 complex did not form in *Wdr11*⁻/⁻ MEFs, indicating that EMX1-GLI3 failed to bind in the absence of WDR11.

D  The relative expression ratios of endogenous GLI3FL/R normalized to the loading control (β-actin) were assessed by Western blot in HEK293 cells after WDR11 overexpression (left panel), in MEFs of WT and *Wdr11*⁻/⁻ embryos (middle panel) and in tissue lysates of WT and *Wdr11*⁻/⁻ kidney (right panel). Band intensities of GLI3FL and GLI3R were quantified by ImageJ.

Source data are available online for this figure.

GLI consensus sites but present a few potential EMX1/2 binding sites (TAATTA), suggesting that *GNRH1* may be a downstream target of Hh-GLI-EMX pathway. We tested these ideas in two immortalized GnRH neuronal cell lines derived from human and mouse embryos, FNCB4-hTERT [28] and GN11 [29], respectively. Note that FNCB4-hTERT expresses *EMX1*, while GN11 expresses *Emx2*, and MEFs do not express GNRH1. Purmorphamine treatment significantly increased the mRNA levels of *GNRH1, WDR11, EMX1/2, PTCH1* and *GLI3,* demonstrating not only the presence of functionally active Hh signalling in the GnRH neuronal system but also identifying *GNRH1* and *WDR11* as novel Hh downstream targets. Notably, *WDR11*-shRNA-infected FNCB4-hTERT or *Wdr11*$^{-/-}$ MEFs contained significantly decreased mRNA levels of *GNRH1* and other Hh target genes (Fig 6A). Moreover, WDR11 KD/KO cells were less capable of responding to purmorphamine in inducing these genes, compared to the controls (Fig 6B), suggesting an important role of WDR11 in mediating Hh pathway activation.

Transfection of GLI1 and GLI3, but not GLI2, induced *EMX1* expression in FNCB4-hTERT, indicating that *EMX1* is a transcriptional target of GLI. However, WDR11 was not induced by the GLI members, suggesting that although WDR11 was inducible by purmorphamine treatment, it was not a direct target of GLI transcription factors (Fig 6C). We next examined the role of WDR11/GLI3/EMX1 in GnRH expression. Transfection of FNCB4-hTERT cells with GLI3FL or EMX1 alone did not have any effects. When EMX1 and GLI3FL were co-transfected, a moderate increase (~30%) of *GNRH1* mRNA was observed, possibly mediated by the endogenous WDR11. When WDR11, EMX1 and GLI3FL were co-transfected, a significant increase (~130%) was observed, suggesting a cooperative activation by the tertiary WDR11-EMX1-GLI3 complex. Since WDR11 could also bind to GLI3R, we tested whether GLI3R would repress GNRH1 expression in this context. We did not observe any inhibitory effect by GLI3R when co-transfected with EMX1 or EMX1/WDR11 (Fig 6D). Based on these results, we postulated that GnRH neurons lacking WDR11 would be less capable of responding to Hh signal and suffer from insufficient GnRH expression. Interestingly, neither purmorphamine nor WDR11 KD altered the random motility of FNCB4-hTERT cells *in vitro*, suggesting the possibility that WDR11-mediated Hh signalling does not regulate the intrinsic motility of GnRH neurons (Appendix Fig S6B).

### WDR11 regulates Hh target gene expression at the chromatin level

To support the mechanistic link between WDR11 and Hh target gene expression, we examined whether WDR11 protein physically associates with the Hh target gene promoter sequence by performing quantitative chromatin immunoprecipitation (qChIP) experiments in HEK293 cells, after overexpression of Myc-tagged WDR11. We show that the *PTCH1* promoter sequence containing the well-established GLI-binding consensus sites could be pulled down with Myc antibody at significantly higher levels, compared to the non-specific IgG or the empty vector-transfected cells (Fig EV2). This result demonstrates that WDR11 is physically involved in GLI-mediated transcriptional regulation of *PTCH1*. In this scenario, WDR11 is likely to provide scaffolding function through a protein–protein interaction, as WDR11 does not contain any known DNA-binding motifs.

### WDR11 modulates GLI3 processing

Ciliary mutants have been implicated in defective GLI processing, resulting in accumulation of either GLI3FL [30–32] or GLI3R [7,8]. Therefore, we examined whether WDR11 regulates the processing of GLI3 protein. WDR11 overexpression in HEK293 cells showed an upregulation of both GLI3FL and GLI3R with an overall increase in FL/R ratio (Fig 5D, left panel). *Wdr11* KO caused an accumulation of Gli3FL in both MEFs and tissue lysates, while the WT exhibited a relative abundance of Gli3R (Fig 5D, middle and right panels), suggesting that *Wdr11* deficiency disturbed Gli3FL cleavage. The full-length GLI proteins produced in ciliary mutants were reported to be non-functional [30]. Indeed, Hh downstream target genes *Ptch1* and *Gli3* were significantly downregulated in *Wdr11*$^{-/-}$ MEFs (Fig 6A).

### A novel WDR11 mutation associates with pituitary dysgenesis, anxiety disorder and childhood obesity

Based on our new understanding of the biological activities of WDR11, we sought to investigate the effects of human mutations and associated phenotypes. We previously reported five missense mutations of *WDR11* in seven CHH/KS patients, all diagnosed with classical CHH/KS phenotypes [13]. Here, we additionally report a novel *WDR11* mutation associated with pituitary dysgenesis, anxiety

---

**Figure 6.  WDR11 cooperates with Hh pathway to regulate gene expression.**

A   Gene expression analyses of GN11 after 48-h treatment with purmorphamine (Pur) or solvent dimethylformamide (DFM) by RT–qPCR. Pur significantly induced *Ptch1, Gli3, GnRH1* and *Wdr11*, while *Emx2* was unresponsive ($P = 0.44$; $t = 0.87$). In FNCB4-hTERT, Pur significantly induced *EMX1, PTCH1, GLI3, GNRH1* and *WDR11* mRNA levels compared to DFM. ShRNA-mediated *WDR11*-targeted knockdown (57%) in FNCB4-hTERT cells or *Wdr11* KO in mouse led to significant reduction in the expression of these genes, compared to the control. Data are shown as means ± SEM from three independent experiments after unpaired Student's *t*-test (*$P < 0.05$; **$P < 0.01$; ***$P < 0.001$; ****$P < 0.0001$).

B   RT–qPCR analyses of FNCB4-hTERT and MEFs with WDR11 KD/KO, respectively. Pur treatment for 48 h caused a significant upregulation of *GLI3* and *EMX1* in control FNCB4-hTERT cells, which was dampened down in WDR11 KD cells ($n = 3$). Similarly, the expression of *Gli3* and *Ptch1* was significantly upregulated by Pur in WT ($n = 7$), but dampened down by *Wdr11* KO MEFs ($n = 5$). Data are presented as means ± SEM from independent experiments after unpaired Student's *t*-test (*$P < 0.05$; **$P < 0.01$; ****$P < 0.0001$).

C   Overexpression of *GLI1* and *GLI3* in FNCB4-hTERT significantly upregulated *EMX1* mRNA compared to empty vector, while WDR11 remained unresponsive. Data represent means ± SEM of three independent experiments with one-way ANOVA followed by Dunnett's *post hoc* test (****$P < 0.0001$).

D   RT–qPCR analyses of *GNRH1* in FNCB4-hTERT cells transfected with WDR11, EMX1, GLI3FL or GLI3R expression constructs. EMX1 or GLI3FL alone showed no effects. Co-transfection of EMX1/GLI3FL caused a moderate increase (37%) of *GNRH1*, while co-transfection of WDR11/EMX1/GLI3FL significantly increased the expression (231%). GLI3R/EMX1 or WDR11/GLI3R/EMX1 did not show any significant response. Data represent means ± SEM of three independent experiments with one-way ANOVA followed by Dunnett's *post hoc* test (**$P < 0.01$; ***$P < 0.001$).

Source data are available online for this figure.

disorder and childhood obesity (Fig 7A, Appendix Fig S7). The index patient demonstrated growth hormone deficiency (treated with GH), obesity, attention-deficit hyperactivity and anxiety disorder. He also manifested delayed puberty and his response to GnRH stimulation was pre-pubertal. The index patient's affected brother had childhood obesity, biochemical growth hormone deficiency and glucose intolerance (see Appendix Text and Table S4 for details). Brain MRI scan revealed a hypoplastic adenohypophysis and an ectopic neurohypophysis in both patients (Fig 7B).

Whole-exome-sequencing analysis of the patient identified a heterozygous rare variant of WDR11 (c.1610C>T, p.Pro537Leu) predicted to be deleterious by two prediction programs, co-segregating with the pituitary phenotype (Appendix Fig S7B). This variant has received an rs number (rs761599645) and is reported in the dbSNP with a frequency of 0.00001647 (2/121,398; Appendix Table S2C). The patients had inherited the mutation from their mother, who had normal brain MRI, suggesting incomplete penetrance of the mutation.

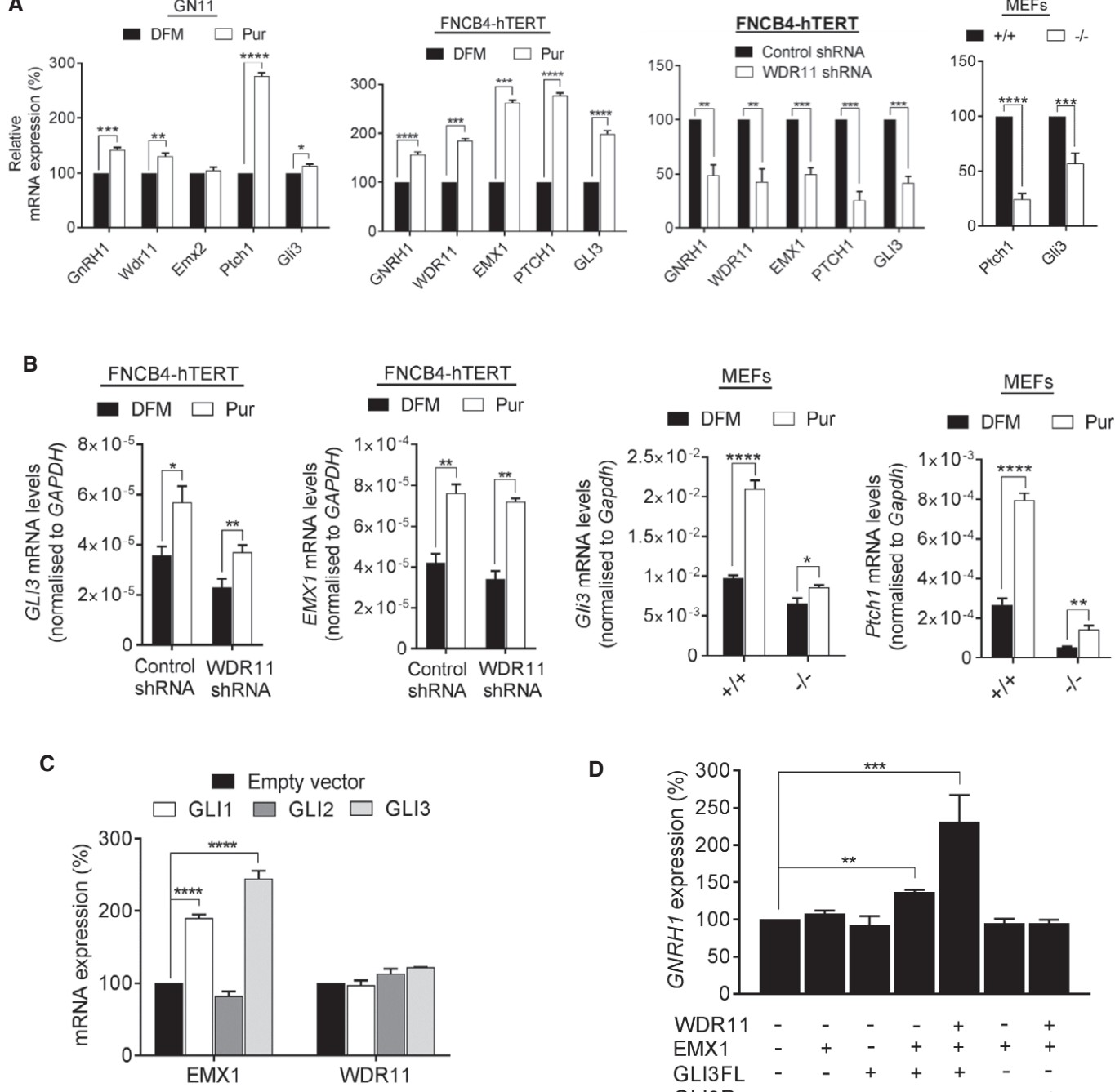

Figure 6.

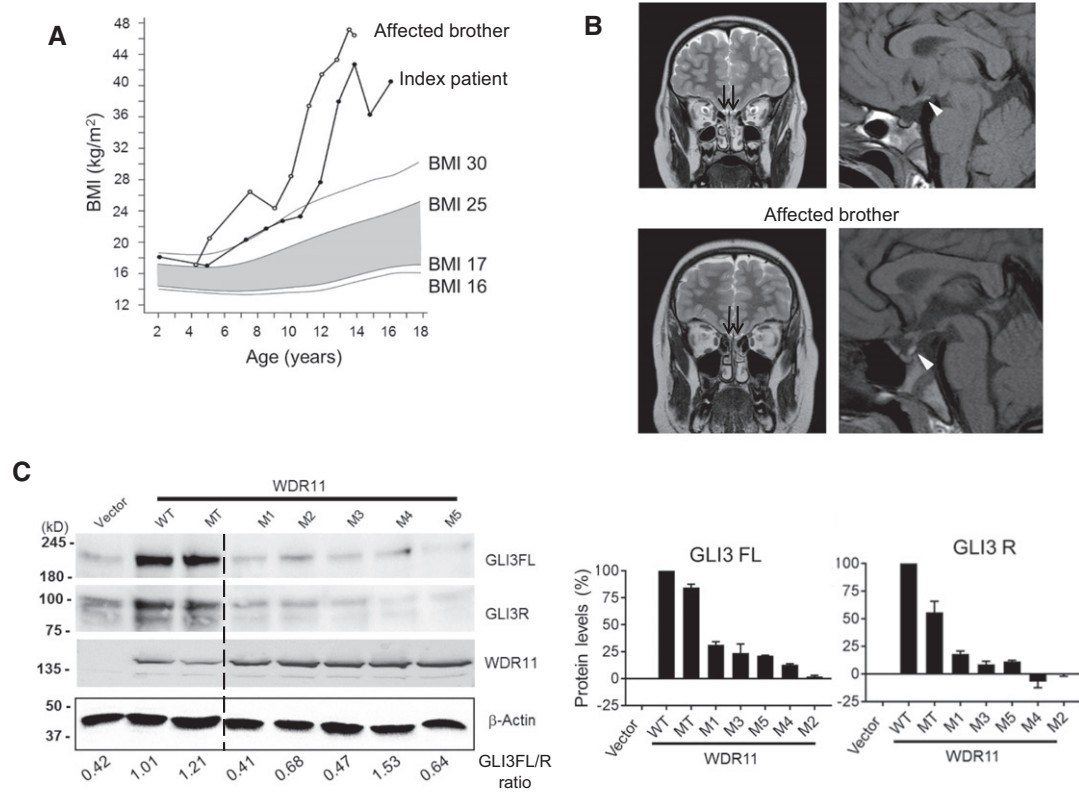

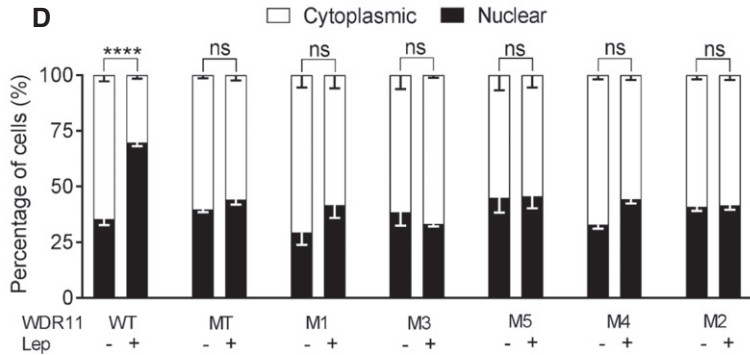

**Figure 7. Disease-associated human WDR11 mutations are functionally defective.**

A   BMI curves of the index patient (filled circles) and his affected brother (open circles). The shaded area represents normal BMI for age (from grade 2 thinness to overweight).

B   MR images of patients. Sagittal T1-weighted image (right panel) of index patient demonstrates truncation of the pituitary stalk with slightly bright ectopic neurohypophysis (white arrowhead) and hypoplasia of the adenohypophysis. Image of his affected brother with same WDR11 mutation demonstrates ectopic neurohypophysis (white arrowhead) next to the normal pituitary stalk. Hypoplasia of the adenohypophysis is also seen. Coronal T2-weighted images (left panel) of both patients demonstrate normal olfactory tracts (black arrows).

C   Western blot of HEK293 cells transfected with WT or various mutants of WDR11 analysed by antibodies against GLI3 (for both FL and R form), Myc epitope (for WDR11) and β-actin loading control, with the relative ratios of GLI3FL/GLI3R shown below (left panel). Band intensities of GLI3FL and GLI3R quantified by ImageJ were normalized to loading control and compared to the WT (right panel). Data are mean ± SEM from three independent experiments.

D   Immunofluorescence images of HEK293 cells transfected with GFP-tagged WT or mutant WDR11 constructs were analysed after leptomycin B (+) or solvent (−) treatment (representative images are shown in Appendix Fig S7C). The percentage of cells showing either nuclear or cytoplasmic localization of WDR11 is plotted. Data from four independent experiments, analysing 100–200 cells in each experiment, are presented as mean ± SEM with two-way ANOVA followed by Tukey's *post hoc* test (****$P < 0.0001$).

Source data are available online for this figure.

## Disease-associated missense mutations of WDR11 cause loss of activities

We investigated the human mutations of WDR11 identified so far—rs201051480 (M1), rs318240760 (M2), rs144440500 (M3), rs318240761 (M4), rs139007744 (M5) and rs761599645 (MT)—in functional assays to confirm their pathogenicity and validate the disease mechanism. First, we tested their ability to increase GLI3 protein levels when overexpressed in HEK293 cells. All mutant WDR11 constructs showed a reduced capacity to increase GLI3FL and GLI3R. MT variant was least functionally affected, fitting with the relatively mild clinical phenotype (Fig 7C). Second, we examined their nuclear localization behaviour. None of the mutants showed a significant shift in nuclear/cytoplasmic distribution when the nuclear export was blocked by leptomycin B, potentially indicating their reduced capacity to enter the nucleus (Fig 7D and Appendix Fig S7C). Third, we compared their ability to form GLI3/EMX1/WDR11 complex. Three of the mutants (MT, M2 and M4)

showed a significant loss of binding with EMX1, thus failing to form the tertiary complex (Fig 5A, right panel).

### Hh agonist rescues GnRH deficiency and ciliogenesis defects

Although the congenital dysmorphic features caused by *WDR11* mutation are not possible to correct, many of the metabolic dysfunctions can be modulated at the molecular level. GnRH neurons have multiple primary cilia where kisspeptin receptors are concentrated, and kisspeptin receptor-dependent GnRH firing at puberty is reduced when ciliogenesis is ablated [19]. The incidence of reversible CHH/KS cases is estimated to be as high as 22% with spontaneous recovery of reproductive function [33], suggesting a possibility that modulation of defective signal pathways may rescue the metabolic abnormalities in these patients. Given the known involvement of the Hh pathway in the development and maintenance of the HPG axis [34] and the importance of primary cilia in both exocrine and endocrine organs [35], we contemplated the

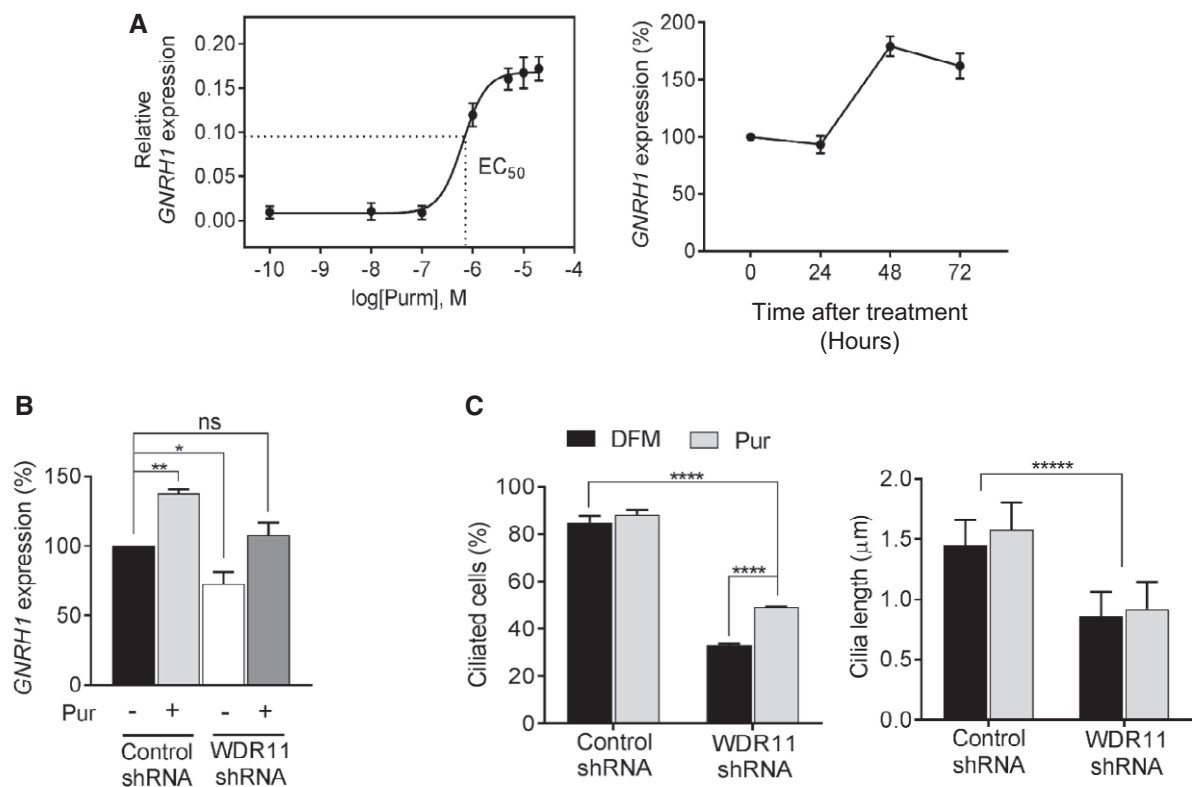

**Figure 8.  The efficacy of Hedgehog agonist in GnRH deficiency.**

A  Purmorphamine dose–response curve showing the induction of *GNRH1* mRNA in FNCB4-hTERT cells with 48 h of treatment ($EC_{50}$ = 675 nM). Time course profile shows Pur can progressively induce GNRH1 mRNA up to 179% peaking at 48 h before dampening down to 162% at 72 h. Error bars are SEM ($n$ = 3), normalized to *GAPDH*.

B  RT–qPCR analysis of *GNRH1* in FNCB4-hTERT cells infected with either control shRNA or *WDR11*-shRNA, after treatments with DFM (−) or 10 μM Pur (+) for 48 h. WDR11 KD caused a reduction of endogenous levels of GNRH1 mRNA, but Pur treatment increased it to the basal level, resulting in no difference from the untreated normal control cells ($P$ = 0.70, $F_{(3,8)}$ = 1.39). Data are means ± SEM of three independent experiments (*$P$ < 0.05; **$P$ < 0.01; one-way ANOVA with Dunnett's *post hoc* test).

C  Immunofluorescence images of FNCB4-hTERT infected with either control shRNA or *WDR11*-shRNA were analysed to compare the percentages of ciliated cells and the length of cilia by staining with acetylated tubulin antibody (representative images are shown in Appendix Fig S9). Treatments with Pur significantly increased the percentages of ciliated cells in the *WDR11* knockdown, but had no effect on the cilia length. Total 300–500 cells were analysed in each group and data are presented as mean ± SEM from three independent experiments using two-way ANOVA followed by Tukey's *post hoc* test (****$P$ < 0.0001; *****$P$ < 0.00001).

Source data are available online for this figure.

therapeutic potential of Hh agonist to reverse GnRH deficiency and possibly other metabolic abnormalities associated with ciliopathy. We first assessed the efficacy of purmorphamine to induce GnRH expression in FNCB4-hTERT cells. There was a dose-dependent increase of *GNRH1* mRNA, exhibiting an EC50 of 675 nM. Purmorphamine also caused a time-dependent induction up to 179% at 48 h, which slightly decreased to 162% at 72 h (Fig 8A). The induction of GnRH at the protein level was also observed in GN11 cells after 48 h of purmorphamine treatment or WDR11 overexpression (Appendix Fig S8). We next explored the possible rescue of GnRH deficiency by purmorphamine. To mimic the state of haploinsufficiency or hemizygosity of *WDR11* in the patients, we knocked down the expression of endogenous WDR11 to 50% in FNCB4-hTERT cells, which also caused 30% reduction in *GNRH1* expression, compared to uninfected control. After 48 h of purmorphamine treatment, *GNRH1* mRNA was increased by ~30% in the WDR11-shRNA-infected cells, reaching levels comparable to that of untreated normal control cells, essentially demonstrating a functional rescue (Fig 8B). Encouraged by this finding, we investigated whether purmorphamine could also rescue defective ciliogenesis in these cells. WDR11 KD caused a decreased frequency of ciliation and shortening of cilia, as similarly observed in *Wdr11*$^{-/-}$ MEFs. Purmorphamine could not increase the cilia length, but increased the percentage of ciliated cells by ~20%, resulting in nearly 50% ciliation. Although still significantly lower than the normal control, the increased ciliation could contribute to the functional rescue and restoration of the signalling capacity (Fig 8C and Appendix Fig S9).

## Discussion

We propose that WDR11 is a scaffolding protein that facilitates the processing and trafficking of GLI3 from the ciliary base to the nucleus, orchestrating the GLI3/EMX1-mediated transcriptional activation of the Hh target genes. EMX1/2 is likely to be a tissue-specific transcriptional activator downstream of GLI1/3. WDR11 is induced by a Hh agonist, but is not a direct target of GLI, and thus, the upstream regulator of WDR11 still remains unclear. Human mutations of WDR11 may cause the disease phenotypes via three mechanisms: (i) defective ciliogenesis compromising the sensitivity to Hh and other cilia-dependent signal pathways; (ii) defective GLI3 processing required for optimal regulation of Hh downstream signal activation; and (iii) disrupted trafficking and formation of GLI3/EMX1/WDR11 complex required for the coordinated and maximum activation of target genes. Considering multiple potential binding partners of WDR11 [13], it is quite possible that WDR11 recruits different tissue-specific regulators in different organs.

CHH/KS have been classically considered as rare hypothalamic diseases caused by the lack of GnRH neuron reactivation at puberty. Our data demonstrate that CHH/KS may be part of the spectrum of other more frequent developmental disorders with common signalling defects. Our WDR11 mouse model recapitulated many of the clinical features of human ciliopathies, some of which overlap with the CHH/KS phenotypes. Embryonic lethality of *Wdr11*-null mutants indicates the lack of redundancy for *Wdr11* function in mouse and underpins its importance especially in early organogenesis. Heterozygous mutations of Hh pathway genes result in a full phenotypic spectrum of HPE in humans, but not in mice, suggesting that

humans may be more sensitive to the dose of Hh signal [36,37], thus explaining why only heterozygote *WDR11* mutations have been found in the patients. Rare heterozygous variants of *GLI2* [38] and *GLI3* [39] have been reported in CHH/KS individuals, but because they coexisted with known CHH/KS gene mutations, it was difficult to confirm their pathogenicity. We propose that the genetic screening of CHH/KS should include the Hh pathway genes and consider the possibility of overlap with a mild form of HPE and other ciliopathies. KS individuals with congenital heart diseases may represent a distinct subgroup with Hh signalling deficiency, as they showed a higher degree of association with short stature, lacked a family history of KS and had intellectual disability [17]. WDR11 may also be responsible for many of the phenotypes associated with 10q26 deletion syndrome which include craniofacial dysmorphism, microcephaly, intellectual disability, cardiac defects, growth retardation, micropenis, cryptorchidism, hearing loss, coloboma and left ventricular enlargement [16,40]. Further investigations employing tissue-specific conditional knockouts in combination with disease-specific mutation knockins are currently in progress to understand the effects of WDR11 mutations.

We demonstrate for the first time a direct regulation of GNRH1 transcription by Hh/GLI signalling cooperatively mediated by WDR11 and EMX1. Other homeobox proteins (VAX1, SIX6 and DLX) have been implicated in GnRH expression, gonadotroph differentiation and pituitary organogenesis [41,42]. EMX1/2 expression defines the dorso-ventral patterning of the forebrain [26,27]. *Emx2* is expressed in the developing urogenital system, and the kidneys, ureters, gonads and genital tracts are completely absent in *Emx2* mutant mice [43]. *Drosophila* homolog *ems* regulates the formation of olfactory circuitry via a POU domain transcription factor *acj6* [44]. Recently, *Emx2* was shown to mediate hair bundle polarity reversal in mechanosensory hair cells [45]. Combined, EMX1/2 may have functions in ciliogenesis or ciliary signal pathways in the reproductive and neurosensory organs.

Our data show that WDR11 deficiency causes childhood obesity, implicating Hh signalling in metabolic diseases. Hh regulates adipocyte differentiation and fat accumulation and controls appetite through leptin receptor trafficking mediated by the primary cilia of hypothalamic neurons [46–48]. Notably, several genome-wide association studies reported *WDR11* locus in association with childhood obesity in Hispanic boys [49], the increase of visceral fat in women [50] and a high fasting cholesterol (triglycerides) level in East Asian individuals [51]. In addition, G-protein-coupled receptors, including somatostatin receptor 3, serotonin receptor 6 and melanin-concentrating hormone receptor 1, localize to cilia and regulate feeding behaviour and energy balance [52]. WDR11 may interact with other ciliary components known to be involved in human ciliopathies such as BBSomes [53] and IFT proteins [54], mutations of which associate with metabolic disorders [35]. Our study may also warrant consideration of pharmacological Hh agonist treatment of CHH/KS and other endocrine disorders caused by ciliary defects. Purmorphamine is a purine derivative that directly binds to and activates SMO [55]. SAG (Smoothened Agonist) is another compound that activates SMO by stabilizing a specific conformation of SMO at $EC_{50} = 3$ nM [56]. The combined use of Hh/SMO agonists with mechanistic differences may be useful in the therapeutic manipulation of Hh signalling.

# Materials and Methods

### Generation of *Wdr11* knockout mouse

The *Wdr11* knockout mouse line Ayu21-KBW205 was generated at the Institute of Resource Development and Analysis, Kumamoto University. The ES cell line KAB6 was electroporated with pU-21W genetrap vector (GenBank Accession number AB427140) which contains a beta-galactosidase/neomycin phosphotransferase fusion gene (beta-geo) with reduced CpG motifs, which allows promoter trapping as previously described [57]. After selection in G418, the genomic DNA from the resistant ES clone was analysed by PCR and Southern blotting to confirm single-copy integration and the insertion site was determined by 5′-RACE. Chimeric mice were derived from the confirmed ES clones using C57BL/6 females and the germ line transmission of the trap allele was verified by Southern blotting of genomic DNA from F1 mouse progenies. The PCR primers used to confirm the vector integration sites and genotyping are schematically shown in Appendix Fig S1A. The primer sequences used are summarized in Appendix Table S1. Genotype-specific PCR was performed on genomic DNA extracted from ear biopsy or amniotic sac.

### Breeding and maintenance of animals

All studies were conducted in accordance with Animal (Scientific Procedures) Act 1986 implemented by the Home Office in England. All mice were housed at the Biological Research Facility at St. George's, University of London under the approved institutional protocols and local regulations following the national guidelines for the care and use of laboratory animals (18–23°C, 12-h light/dark cycle, normal chow diet). Transgenic mice were back-crossed onto a C57BL/6 background for six generations prior to data collection. Animals were randomly assigned for studies using computerized random order generator (GraphPad, QuickCalcs). Researchers were blinded to genotype while performing all analyses.

### Analysis of growth, reproductive function, obesity and skeletal system of mouse

Mouse embryos of defined gestational ages were obtained by timed mating of heterozygote male and female mice, with an observed vaginal plug counted as 0.5 days post-coitum. Body weights were recorded weekly for 10 weeks using a calibrated electronic balance. Body lengths were measured from the nose to the base of tail of the mice using a tape ruler. Male and female mice were examined daily from 2 weeks of age until the day of preputial separation and vaginal opening, respectively. For males, anogenital distance was recorded every 3–7 day. To assess the oestrous cycle in females, vaginal smears were obtained daily for 3 weeks and stained and identified as described [58]. To assess fertility of the male mice, the numbers of pups per litter produced after mating with WT B6 females were tracked for a period of 7 months.

For cartilage and bone analysis, alcian blue and alizarin red staining was performed according to the standard protocols using a staining cocktail (1 part 0.3% alcian blue in 70% EtOH, 1 part 0.1% alizarin red in 95% EtOH, 1 part 100% acetic acid and 17 part 100% EtOH) for 4–10 days. Surrounding tissue was cleared by immersion

in 1% KOH followed by gradual transfers starting from 1% KOH/glycerol and finally to 100% glycerol. For histopathology analysis of fatty liver, Oil Red O-stained liver specimens were evaluated by light microscopy.

### Immunohistochemistry, histological analysis and X-gal staining

Dissected embryos or organs were fixed in Bouin's fixative or 4% paraformaldehyde prior to paraffin embedding. Sections (4–10 μm thick) were cut using an electric microtome (Leica RM2255), mounted on poly-L-lysine-coated microscope slides, de-paraffinized with Histoclear (National Diagnostics) and rehydrated in PBS. Following antigen retrieval in sodium citrate buffer (10 mM sodium citrate, 0.05% Tween-20, pH 6.0), the sections were incubated in a blocking buffer (5% skim milk, 2% goat serum, 0.2% Triton X-100 in PBS), probed with the primary antibody diluted in the blocking buffer, washed in PBS, followed by the addition of the appropriate secondary antibody diluted in blocking buffer for 30 min. After counterstaining with DAPI, the slides were mounted with Mowiol 4–88 (Fluka) and examined using Zeiss Axiovert 200 M Inverted or Zeiss Axioplan 2 Upright. Haematoxylin/eosin (H/E) staining was performed on paraffin sections according to standard protocols.

For whole-mount lacZ staining, isolated embryos were fixed in a fixative solution (0.2% glutaraldehyde, 2% paraformaldehyde, 5 mM EGTA, 2 mM MgCl$_2$ in PBS), washed in PBS and stained overnight at 37°C with a developing solution (40 mg/ml X-gal, 200 mM K$_3$Fe(CN)$_6$, 200 mM K$_4$Fe(CN)$_6$, 1 M MgCl$_2$, in PBS). For histological analyses, the X-gal stained embryos were post-fixed in 4% paraformaldehyde, dehydrated and embedded in paraffin, which were then sectioned at 6–8 μm thickness to be counter-stained with eosin.

### Electron microscopy

The paraffin-embedded sections of 12-week-old mouse brain were re-processed for the purpose of electron microscopy as previously described [59] and embedded in Agar 100 resin (Agar Scientific Ltd). Thin sections were cut on a Reichert-Jung Ultracut E ultramicrotome and placed on uncoated copper grids, which were stained with 2% aqueous uranyl acetate for 45 min and Reynold's lead citrate for 1.5 min [60]. Stained sections were analysed under a Hitachi H-7100 transmission electron microscope.

### Immunocytochemistry

Cells grown on glass coverslips were fixed with 4% paraformaldehyde, permeabilized with 0.2% Triton X-100 in PBS, incubated in blocking buffer (2% heat-inactivated goat serum, 0.2% Triton X-100 in PBS) and probed with primary antibodies diluted in blocking buffer. After washing, appropriate secondary antibodies were added along with DAPI, before mounting in Mowiol. Cilia formation was induced by serum starvation for 24–48 h and visualized by anti-acetylated tubulin, anti-Arl13b or anti-gamma-tubulin antibodies. In some cases, cells were pre-treated with different drugs for 10 h before staining. For analysis of the sperm, caudal epididymides from sexually mature males were finely minced and the sperm released into the buffer was collected and mounted. The slides were then air-dried, fixed in 1.5%

paraformaldehyde (v/v in PBS), permeabilized using 0.2% Triton X-100, blocked in 2% BSA and stained with primary antibodies and DAPI. Fluorescence microscopy was performed using Zeiss Axioplan 2 Upright or Zeiss Confocal LSM 510 and analysed by using ImageJ software (https://imagej.nih.gov/ij/index.html).

### Antibodies, Western blot and co-immunoprecipitation assays

Primary antibodies used were raised against GnRH1 (Abcam ab5617), prolactin (Santa Cruz sc-46698), WDR11 (Abcam ab175256; Santa Cruz sc-163523), GLI3 (Novus Biologicals AF3690), EMX1/2 (Invitrogen PA5-35373), acetylated alpha-tubulin (Sigma T6793), β-actin (Santa Cruz sc-1615), gamma-tubulin (Sigma T6557 and GTU88), Arl13b (Proteintech 17711-1-AP), Myc (M4439, Sigma-Aldrich), GFP (Abcam ab290) and beta-GAL (Abcam ab9361). Secondary antibodies (all from Molecular Probes) were conjugated with Alexa Fluor 555 (goat anti-rabbit IgG), Alexa Fluor 568 (goat anti-mouse IgG), Alexa Fluor 488 (goat anti-rabbit and donkey anti-goat IgG) and used at 1:5,000 dilution. For negative controls, non-specific IgG (Sigma-Aldrich) was used instead of primary antibodies.

Total cell lysates were extracted in 1% Triton X lysis buffer containing a protease inhibitor cocktail (Sigma, P8340) and a phosphatase inhibitor tablet (Sigma, P5726). Protein concentration was determined by the Bradford assay (Bio-Rad). SDS–PAGE was carried out as previously described [13]. Proteins were transferred onto Hybond-ECL membrane (Amersham), blocked in blocking buffer [5% skim milk in TBS with 0.05% Tween-20 (TBST)] and incubated overnight at 4°C with the appropriate primary antibody diluted in blocking buffer. Membranes were washed with TBST, probed with the secondary antibody conjugated with horseradish peroxidase and visualized by enhanced chemiluminescence (GE Healthcare). Band intensities were quantified by ImageJ and normalized to β-actin. Co-immunoprecipitation was performed using the lysis buffer (50 mM HEPES, 150 mM NaCl, 10% glycerol, 1% Nonidet P-40 and 1 mM EDTA) containing protease and phosphatase inhibitors. The pre-cleared lysate was precipitated with anti-Myc antibody and the immune complexes captured on protein A/G-agarose beads (Santa Cruz Biotechnology) were washed with the lysis buffer and analysed by SDS–PAGE and Western blotting.

### Quantitative chromatin immunoprecipitation

Quantitative chromatin immunoprecipitation assay was performed as previously described [61,62] with modification. Briefly, HEK293 cells transfected with 6xMyc-tagged WDR11 in pDEST or empty vector were cross-linked with 1% formaldehyde for 15 min and inactivated by adding 125 mM glycine. Cells were washed and resuspended in lysis buffer (1% SDS, 10 mM EDTA 10 mM HEPES (pH 7.9), 0.5 mM PMSF), before being sonicated to fragment chromosomal DNA which was then diluted 10-fold into IP buffer (0.01% SDS, 1.1% Triton X-100, 1.2 mM EDTA, 16.7 mM Tris pH 8.1, 167 mM NaCl and protease inhibitors). The pre-cleared lysate was incubated with anti-Myc antibody or mouse IgG with protein A/G-agarose beads (Santa Cruz Biotechnology) overnight at 4°C. The immune complexes were eluted with an elution buffer (0.5% SDS, 5 mM EDTA, 25 mM Tris–HCl pH 7.5,

20 mM NaCl) and incubated at 65°C overnight. The DNA sample recovered by QIAquick PCR purification kit (QIAGEN) was analysed by qPCR using primers flanking the putative Gli-binding sites (GliBS-A and GliBS-B) [63] within PTCH1 promoter sequences. The data were normalized to the input DNA which was collected before the immunoprecipitation.

### Cell culture

HEK293 and NIH3T3 authenticated by isoenzyme (interspecies) and STR (intraspecies) analysis were purchased from American Type Culture Collection (ATCC). HEK293T cells were maintained in DMEM containing 2 mM L-glutamine and 100 µg/ml penicillin/streptomycin (Sigma-Aldrich), supplemented with 10% foetal bovine serum (FBS). NIH3T3 cells were cultured in DMEM supplemented with 10% newborn calf serum (NCS). GN11 cells (obtained from Dr Sally Radovick) [64] were cultured in DMEM with 7% FBS and 3% NCS. FNCB4-hTERT cells were grown in F12 Coon's modification (Sigma-Aldrich F6636) with 7% FBS. FNCB4-hTERT, a human embryonic olfactory GnRH neuroblast cell line isolated from olfactory epithelium of 8- to 12-week-old embryo, has been immortalized by introducing human telomerase via pBabe-hTERT retroviral infection, as previously described [28]. All cells were grown in a humidified 37°C incubator with 5% $CO_2$ and routinely tested for mycoplasma contamination using MycoTOOL kit (Roche). All experiments were performed at passages < 25–30.

For generation of MEFs, E12.5 mouse embryos were dissected, and the head and visceral organs were removed. After digestion with trypsin–EDTA (Sigma T3924), samples were homogenized and plated onto a 6-cm tissue culture dish containing DMEM supplemented with 10% FBS. When the cells became confluent, they were passaged regularly by 1:2 split. All analyses were performed on MEFs at passages between 3 and 6.

Compounds used for treatments of cells were purmorphamine (10–40 µM, Sigma-Aldrich), cyclopamine (40 µM, Selleck), GANT61 (40 µM, Abcam) and dimethylformamide (10 µM, Sigma-Aldrich). Plasmid DNAs were transfected using Fugene (Promega) or Polyfect (Qiagen).

### Motility assay

Non-directional motility was measured by time-lapse live imaging using an Olympus IX70 inverted microscope (Hamamatsu C4742–95, Hamamatsu, Japan) at 37°C with 5% $CO_2$. Cells were serum-starved for 24 h before treatments with DFM or purmorphamine for 18 h, and cell images captured every 15 min for 20 h were analysed using ImagePro Plus to track the movement (Media Cybernetics, Bethesda, MD). Twenty random cells were tracked in 3–5 independent repeat experiments.

### Lentiviral shRNA infection

Sequence-verified shRNA targeting human WDR11 (Sigma-Aldrich Mission, SHCLNG NM_018117) or non-targeting shRNA (Sigma-Aldrich Mission, SHC001) in pLKO.1-puro lentiviral vector was produced in 293FT packaging cells that were co-transfected with Gag-Pol (pLP1), Rev (pLP2) and vesicular stomatitis virus G protein (pLP/VSVG) expression vectors (Sigma-Aldrich). The filtered viral

supernatants were added to the FNCB4-hTERT cells in the presence of polybrene (Sigma-Aldrich).

## Quantitative RT–PCR

Total RNA was extracted from cultured cells or snap-frozen tissues using the RNAeasy kit (QIAGEN) and reverse-transcribed using NanoScript 2 Reverse Transcription kit (Primer Design). The cDNA was amplified using a Maxima SYBR Green qPCR Master Mix (Thermo Scientific) on a Light Cycler 2.0 (Roche). Data were analysed using the $2^{-\Delta\Delta C_T}$ method, normalized to *GAPDH*. The primers used are summarized in Appendix Table S1.

### *In vitro* mutagenesis and plasmids

The full-length WDR11 expression constructs were made in pcDNA-GFP, pcDEST-Myc or pFN21A vector, as previously described [13]. Site-directed mutagenesis was performed using Q5 mutagenesis kit according to the manufacturer's protocol (New England Biolabs). Primer sequences used for the creation and confirmation of the mutant WDR11 constructs are shown in Appendix Table S2. The full-length human GLI 1/2/3 expression constructs in pcDNA vector were obtained from Daewon Kim. The human GLI3R expression construct with 6xMyc epitope in pIRES vector was a gift from James Briscoe. The haemagglutinin (HA)-tagged EMX1 expression construct in pSG5 vector was previously described [13].

### High-resolution episcopic microscopy

For HREM examination, embryo was fixed and processed as described [65]. This technique uses block-face fluorescence imaging and serial sectioning to produce image sequences that are perfectly in register and can be digitally 3D reconstructed. HREM image data sets were optimized and sub-scaled for 3D volume rendering using Osirix MD (www.osirix-viewer.com).

### Zebrafish

Zebrafish wild-type strains (AB × Tup LF) were maintained and staged as previously described [66]. Gene knockdown was performed using antisense morpholino (MO) oligonucleotides (Gene-tools, LLC) designed against the exon 3–intron 3 (5′-GATGGTG ATCTTTTTCTTACCCTGA-3′) and exon 9–intron 9 (5′-TGGGT GGACGGCTATCTTACCAATC-3′) splice sites. Embryos were injected at the 1- to 4-cell stages with 2 ng of MO and incubated at 28.5°C until the desired stage. RT–PCR was performed to validate MO specificity as previously described [67] using primers P1 (5′-TGAAATGGTCCAGAGAAAACTACCA-3′) and P2 (5′-CCAGATTA GAGGGCTCAAAGGGGT-3′) for exon 3 to exon 4 amplicons and P3 (5′-CGTCCCTATCGAGTGGTCAT-3′) and P4 (5′-CAGCCAGTAAGGG CTGGTAG-3′) for exon 9 to exon 10 amplicons. *B-actin* (forward, 5′-AATCCCAAAGCCAACAGAGAGAAGA-3′; reverse, 5′-CTCGTGGA TACCGCAAGATTCCATA-3′) and *gapdh* (forward, 5′-TTAAGGCA GAAGGCGGCAAA-3′; reverse, 5′-AAGGAGCCAGGCAGTTGGTG-3′) primers were used as internal controls. Immunofluorescence was carried out using primary antibodies at 1:200–500 dilutions with appropriate Alexa secondary antibodies used at 1:1,000. *In situ* hybridization was performed using standard protocols with probes

for *sox10* and *ptc1* as described [68]. Head cartilage was visualized using alcian blue staining as described [67]. Live fish were analysed using a Nikon SMZ1000 dissecting microscope with a C-POL polarizing attachment.

### Genetic and clinical analysis of patients

Informed consent from all subjects to publish were obtained and approved by the Ethics Committee of Helsinki University Central Hospital and Medical College of Georgia, Augusta University. The experiments conformed to the principles set out in the WMA Declaration of Helsinki and the Department of Health and Human Services Belmont Report. For the genetic analysis of *WDR11* mutation of patient MT, DNA of the index patient and his family members (Appendix Fig S7A and B) was extracted from saliva using OraGene DNA (OG-500) kit (DNA Genotek Inc., Ottawa, ON, Canada). The whole-exome sequencing of the index patient was performed in the Beijing Genomics Institute (Shenzhen, China) using SureSelect All Exon V4 kit (Agilent, Santa Clara, CA, USA) and 100-bp paired-end sequencing with Illumina HiSeq 4000 technology. The data were filtered to include only novel (not reported in 1,000 Genomes, dbSNP build 141 or ExAC databases) non-synonymous or splice site variants. The segregation of the 13 single nucleotide variants (in genes *WDR11, KIAA1755, EPB41L4A, EML6, AGAP1, GNAI2, SLC6A7, COL28A1, CA3, TAF1L, ARHGAP1, ITM2C* and *CLDN4*) remaining after the filtering was checked by Sanger sequencing (Applied Biosystems ABI3730XL DNA Analyzer) from the patient, his parents and his affected brother. Another two variants fulfilling the filtering criteria, located in *ANKRD36* and *GALNT2*, were predicted to be of low impact and thus not likely to be the causal variant. The number of indel variants was too large to screen systematically by Sanger sequencing, so they were prioritized based on the known or predicted function of the protein product. More specifically, we searched for predicted high impact variants in genes with an associated Gene Ontology (GO) term related to the development or function of the pituitary, forebrain development or ciliary function.

MR studies were performed with a 1.5-T system. Imaging protocol of the hypophyseal area included sagittal turbo spin-echo (TSE) T1-weighted sequences (550/10 [TR/TE], 3 mm thick) with and without the use of IV administered contrast material and 3-mm-thick coronal spin-echo T1-weighted sequences without (475/15 [TR/TE]) and with (525/15 [TR/TE]) contrast material. Contrast-enhanced images were obtained with fat suppression. Coronal and axial 3-mm-thick T2-weighted TSE and axial 4-mm-thick FLAIR sequences with long TR of the whole brain were also obtained. The images were evaluated by a neuroradiologist for the appearance of the olfactory tracts and the hypophyseal area or other brain abnormalities.

### Statistical analysis

For analyses of groups with unequal variances, *F*-test was employed. Statistical significance was determined by unpaired two-tailed Student's *t*-test, one-way ANOVA followed by Dunnett's *post hoc* test and two-way ANOVA followed by Bonferroni's or Tukey's *post hoc* test for independent experiments. Data were analysed using GraphPad Prism 6 Software (La Jolla, USA).

## Data availability

All data generated or analysed during this study are included in this published article and its supplementary information files. WDR11 sequence variants are deposited in the dbSNP (Appendix Table S2C).

**Expanded View** for this article is available online.

## Acknowledgements

We thank Maria McGlynn and Yvette Bland for technical assistance, and Helenä Kääriäinen, Nicola Ragge and Moira Blyth for discussions related to clinical findings. YJK, JYL, LCL and S-HK are supported by Medical Research Council (MRC) grant MR/L020378/1. LCL is also supported by NIH grant HD33004. MA and KA are supported by JSPS KAKENHI grant (JP17017031, JP20300146). TM is supported by the MRC grant (U117562103) and the Francis Crick Institute which receives its core funding from Cancer Research UK, the MRC, and the Wellcome Trust (FC001157, FC001117). H-GK is supported by Carrick Foundation. TR is supported by the Sigrid Juselius Foundation, Foundation for Pediatric Research and HUCH Research Funds. C-HK was supported by a National Research Foundation of Korea (2014R1A2A1A11053562).

## Author contributions

Conceptualization, funding acquisition and supervision: S-HK, LCL; Original draft writing: S-HK; Methodology (creation of models and design of methods): MA, KA, DPSO, NAB, C-HK; Investigation (performing the experiments): Y-JK, J-YL, DPSO, TM, JK, NB, FM, PA, H-TK; Resources (patients): TR, LCL; Manuscript review and editing: DPSO, JPM-B, PA, LCL, NAB, H-GK, TR.

## Conflict of interest

The authors declare that they have no conflict of interest.

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
