## [Review Process File · EMBO Reports]

WDR11-mediated Hedgehog signalling defects underlie a new ciliopathy related to Kallmann syndrome

Yeon-Joo Kim, Daniel P.S. Osborn, Ji-Young Lee, Masatake Araki, Kimi Araki, Timothy Mohun, Johanna Käsäkoski, Nina Brandstack, Hyun-Taek Kim, Francesc Miralles, Cheol-Hee Kim, Nigel A. Brown, Hyung-Goo Kim, Juan Pedro Martinez-Barbera, Paris Ataliotis, Taneli Raivio, Lawrence C. Layman, Soo-Hyun Kim

Review timeline:

Submission date:	14 June 2017
Editorial Decision:	12 July 2017
Revision received:	23 September 2017
Editorial Decision:	8 November 2017
Revision received:	14 November 2017
Accepted:	17 November 2017

Transaction Report:

1st Editorial Decision

12 July 2017

Thank you for the submission of your research manuscript to our journal. We have now received the full set of referee reports that is copied below.

As you will see, the referees acknowledge the potential interest of the findings, but they all have a number of suggestions for how the study should be strengthened, which should be addressed. Referee 1 notes that the mechanistic link between WDR11 and Hh target gene expression should be strengthened and suggests to perform ChIP experiments. Upon further discussion with the referee we suggest to verify the presence of the WDR11/EMX/GLI3 complex on selected target genes like Ptc or Gli1 by ChIP-qPCR.

Given these constructive comments, we would like to invite you to revise your manuscript with the understanding that the referee concerns (as detailed above and in their reports) must be fully addressed and their suggestions taken on board. Please address all referee concerns in a complete point-by-point response. Acceptance of the manuscript will depend on a positive outcome of a second round of review. It is EMBO reports policy to allow a single round of revision only and acceptance or rejection of the manuscript will therefore depend on the completeness of your responses included in the next, final version of the manuscript.

Revised manuscripts should be submitted within three months of a request for revision; they will otherwise be treated as new submissions. Please contact us if a 3-months time frame is not sufficient for the revisions so that we can discuss the revisions further.

Supplementary/additional data: The Expanded View format, which will be displayed in the main HTML of the paper in a collapsible format, has replaced the Supplementary information. You can submit up to 5 images as Expanded View. Please follow the nomenclature Figure EV1, Figure EV2 etc. The figure legend for these should be included in the main manuscript document file in a section called Expanded View Figure Legends after the main Figure Legends section. Additional Supplementary material should be supplied as a single pdf labeled Appendix. The Appendix includes a table of content on the first page, all figures and their legends. Please follow the nomenclature Appendix Figure Sx throughout the text and also label the figures according to this nomenclature. For more details please refer to our guide to authors.

Regarding data quantification, please ensure to specify the number "n" for how many experiments were performed, the bars and error bars (e.g. SEM, SD) and the test used to calculate p-values in all respective figure legends.

We now strongly encourage the publication of original source data with the aim of making primary data more accessible and transparent to the reader. The source data will be published in a separate source data file online along with the accepted manuscript and will be linked to the relevant figure. If you would like to use this opportunity, please submit the source data (for example scans of entire gels or blots, data points of graphs in an excel sheet, additional images, etc.) of your key experiments together with the revised manuscript. Please include size markers for scans of entire gels, label the scans with figure and panel number, and send one PDF file per figure or per figure panel.

I look forward to seeing a revised version of your manuscript when it is ready. Please let me know if you have questions or comments regarding the revision.

REFEREE REPORTS

Referee #1:

Kim YJ et al. report a new and extensive characterization of the mouse and zebrafish WDR11 KO / KD phenotypes. They claim similarity between WDR11 KO/KD and ciliopathies or KS/CHH syndromes as they find shared phenotypic characteristics (hypogonadism, obesity, eye anomalies, syndactyly, heart malformations,...), ciliary deficiency and WDR11/cilia colocalization. Using in vitro experiments, they show the formation of a tertiary complex composed of Gli3-WDR11-EMX and claim WDR11 to enhance Hh-induced expression of GnRH. Importantly, they report a new WDR11 mutation in human leading to defective nuclear localization and EMX binding causing both pituitary dysgenesis and obesity. Finally they show preliminary in vitro data suggesting the use of Hh agonists to rescue WDR11 deficiency.

While we find the paper contains extensive characterization of the WDR11 KO/KD with direct clinical relevance to human disease, we feel that the manuscript cannot be published in the present form as a solidified mechanistic link between WDR11 and Hh pathway remains lacking. For clarity, we would suggest the authors focus on the role of WDR11 in Gli3/EMX complex in chromatin regulation/Hh target gene expression (through WDR11 ChIP experiment for example). Also Immunostaining quality needs to be improved and interpretation of some of the figures could be made easier.

Major comments :

- title should be changed as WDR11 was not shown to directly disrupt Hh signaling in the manuscript

- WDR11 KO validation at the protein level is performed on kidney tissue (Fig.1A). As the paper's main focus is on ME and OB, we would suggest to validate WDR11 KO in brain tissue.

-Overall there is inconsistency in cilia staining ; In general, it is best to use both axonemal and basal body co-stainings when one counts cilia. For key quantitative figures, please include at least one representative image with both stained.

- authors show WDR11-dependency of Hh pathway target gene expression (*ptch1* and *Gli3*) ; however WDR11 cytoplasmic localization upon Hh pathway activation (fig 3D and suppl fig 5A) does not support the functional role of WDR11 binding to nuclear EMX (Fig 5). Rather, it supports the opposite-that WDR11 inhibits EMX binding to *Gli3*. Please clarify.

- The authors claim that WDR^{-/-} mice present obesity, a hallmark of BBS-type ciliopathies. However, in the actual data (suppl. fig 3A). The data shows they have lower weight, a seeming contradiction that weakens the authors ciliopathy claim. This contradiction needs to be resolved.

Minor comments :

- For clarity, we would suggest to report and discuss anatomic considerations regarding pituitary gland (shown in Fig 2F) and pituitary functionality (Fig 2D) before potential consequences (delayed puberty, reproductive dysfunctions, obesity) (discussed in Fig.2A-B-C-G)

- Pituitary gland functionality (GH, PRL, LH, FSH) was done at the mRNA level in the WDR11 KO mice (Fig.2D) ; we would suggest to confirm these results at the protein level, given previous demonstration of post transcriptional regulation.

- Although the authors downplay the result, the finding of reduced GnRH neurons in WDR11 KO brains is an important finding ! Please provide an illustrating figure (fig 2E) and include a discussion of its significance.

- The authors claim fatty livers in WDR^{+/-} mice, although liver histology is not shown

- in Fig.3, the authors claim the colocalization of WDR11 and cilia in OB and ME ; cilia deficiency is however reported in the choroid plexus ; for consistency, we suggest the authors show cilia deficiency in OB and ME

- in fig. 3a WDR11 staining looks cytoplasmic whereas it is nuclear in suppl. fig. 2a ; please comment on this surprising difference if representative.

- Suppl. fig 2b would benefit from a higher magnification picture ; why did the authors use WDR^{+/-} instead of WDR11^{+/+} ? Is there a difference in heterozygotes versus wildtype ?

- in vitro WDR11/cilia colocalization is shown using exogenous WDR11 expression in 3T3. It would be more appropriate to use cell lines spontaneously expressing WDR11 (as GnRH neuronal cell lines)

- for consistency with the murine WDR11 KO, we would suggest to test for WDR11 expression in zebrafish OB, ME and pronephros, as well as potential colocalization with cilia (fig 4G-H)

- fig 4E-F-G-H show morpholino E3I3 phenotype ; for consistency we would suggest to show E9I9 phenotype too (supplemental data)

- GnRH expression upon Hh pathway activation in GnRH neuronal cell lines should be confirmed at the protein level (Fig. 6)

- authors show that WDR 11 is sufficient to enhance Hh pathway-induced GnRH expression in the presence of *Gli3/EMX* ; WDR11 necessity for Hh pathway-induced GnRH expression in the presence of *Gli3/EMX* should be tested (using either WDR^{-/-} MEF or WDR11KD in GnRH neuronal cell lines)

- both the mother and 2 brothers are heterozygous for the WDR11 described mutation ; the mother however does not present any phenotype. Please comment on disease penetrance.

- as previously mentioned, GnRh induction upon Hh pathway activation should be confirmed at the protein level in Fig.8A. Please indicate what are the cells used in the experiment

- authors claim WDR11 expression in the « GnRH migratory niche including nasal cavity and cribriform plate », this is not shown in fig. 1B. The authors should either show the data or remove the reference in the paper.
- fig 3A low quality pictures
- fig 3C would benefit from a lower magnification (more than 1 cell)
- fig 5D : determined instead of « determiend »
- scale bars are missing in various figures, or figure legends
- error bars represent either SEM or SD ; this should be uniformed in the whole manuscript
- suppl fig. 3A, WDR+/- are in the legend, but not in the figure

Referee #2:

This is an interesting paper with solid data on the mouse KO and fish morphants. The links with HPE, cilia function in Hh signaling and Gli3 isoform formation are intriguing and worth publishing in EMBO Reports.

The last needs cleaner Westerns and full size blots, instead of cutting the windows of interest (e.g. Fig. 7A).

The presentation of the can be improved. The text, figures etc need revising. Citations of the figures in the text appear not to be always consistent.

The figures and/or the text must be organized better. Figures do not appear in order along the text. For instance, Figure 3A is named first than figure 2. Figure 4 is before than Figure 3D-E.

Unlike what is stated in the text, Figure 1B does not show a detailed analysis of the brain.

Figure 3A. I do not see the correspondence between the full image and the zoomed region.

Referee #3:

Summary:

1. Does this manuscript report a single key finding?

YES. It reports that, in vertebrate development, the scaffold protein WDR11 is required for ciliogenesis through the Hedgehog signaling pathway, based on knock-out mice and zebrafishes and cellular and molecular analysis.

2. Is the reported work of significance?

YES, since the generation and the analysis of knock-out animals are original. But the importance of WDR11 in development of mice and human has already been documented using other methods, such as the discovery of human genetic mutations.

3. Is it of general interest to the molecular biology community?

YES. It is important to elucidate the function of the protein WDR11.

4. Is the single major finding robustly documented using independent lines of experimental evidence?

YES. The analysis of the knock-out animals is well-performed although the number of animals analyzed seems a little weak (5 or below in each group, when indicated).

The manuscript by Kim et al. analyzes the functional role of the protein WDR11, a WD protein that interacts with transcription factor EMX1. Recently, this protein was found to be mutated in congenital hypogonadotropic hypogonadism (CHH) such as Kallmann syndrome (KS). This protein contains twelve WD domains that form two beta propellers potentially serving as molecular scaffolds. But the role of WDR11 in development remains unclear. Here, the authors report that WDR11 modulates the Hedgehog (Hh) signaling pathway and is essential for ciliogenesis. They have generated both knockout (KO) mice and zebrafishes to investigate the function of WDR11 in vertebrate development and ciliogenesis. After a deep anatomical characterization of the Wdr11 KO mice and KO fishes and a cellular analysis of the WDR11, they have concluded for an important role of Wdr11 in Hh signaling and ciliogenesis. Then by co-immunoprecipitation experiments from transfected HEK-293 and Wdr11 KO MEF cells, they have brought evidence that WDR11 forms a tertiary complex with EMX1 and GLI3 transcription factors. The WDR11 was essential for EMX1-GLI3 interaction. In GnRH neuronal cell lines, the authors propose that WDR11 mediates the expression of these two transcription factors as well as GNRH1 through the activation of the Hh signaling pathways by the agonist purmorphamine. In these cell lines, they also show that co-expression of Gli1 and Gli3 proteins, but not Gli2, induced EMX1 expression (but not of WDR11) indicating that the transcription factors GLI target EMX1 but not WDR11. In addition, they have identified one Kallmann patient with a new genetic mutation in WDR11 (heterozygous). They have analyzed the functional consequence of this mutation in WDR11 together with other genetic mutations previously reported in this protein. All these mutants impaired WDR11 function, such as its translocation to the nucleus and the formation of the ternary complex with EMX1 and GLI3. Finally, they have tested the effect of purmorphamine in rescuing the expression of GNRH1 in GnRH neuronal cell lines treated with WDR11 shRNA.

Many results are of interest and novel. The manuscript reports that, in vertebrate development, the scaffold protein WDR11 is required for ciliogenesis through the Hedgehog signaling pathway, based on knock-out mice and zebrafishes and cellular and molecular analysis. The generation and the analysis of knock-out animals is original. But the importance of WDR11 in development of mice and human has already been documented using other methods. The results are appropriately discussed in the context of earlier literature. They will be of interest for investigators working in field of hypothalamus-pituitary development and more generally in the biology of cilia. WDR11 seems to be a key scaffold protein in these processes, and knock-out animal is a key tool to understand the function of this protein. In general, the experimental data are of sufficient quality to justify the conclusions, and the manuscript is clearly written.

Specific comments:

- The abstract is confusing and some part should be deleted and rephrased. They should remove the part "that are affected not only in CHH/KS...leads to obesity both in mice and men". Indeed, it's confusing to mix the results of the study with interpretation of animal and human phenotype. Also, the authors should remove the text "as one of key links between reproduction and metabolism" and conclude more about their results.
- The authors do not clearly state how many animals (both mice and fishes) were used in the experiments. For example, they used a small number (3 to 5) of Wdr11-null males (see page 6). It would be better to use in most cases at least 5 animals. It should be clarified.
- Figure 2, panel D: could you explain the large differences between males and females for GnRHR, LH and FSH expression in the KO animals?
- Co-immunoprecipitation was done with recombinant WDR11, EMX1 and GLI3 after cell transfections. However, primary antibodies against these proteins have been used (see Materials and Methods). So why the authors did not show co-immunoprecipitation experiments with the endogenous proteins? Would it be possible to do some of these experiments, since such data will reinforce the existence of the ternary complex proposed in native conditions?
- Title Figure 8: remove "therapeutic" since it is not adapted here.
- How selective the drug purmorphamine is in term of drug target due to its low potency (micromolar, see Fig. 8A)? It would be an information useful for the readers outside of the field of Hh signaling. Do better agonists are existing?

Referee #1:

Kim YJ et al. report a new and extensive characterization of the mouse and zebrafish WDR11 KO/KD phenotypes. They claim similarity between WDR11 KO/KD and ciliopathies or KS/CHH syndromes as they find shared phenotypic characteristics (hypogonadism, obesity, eye anomalies, syndactyly, heart malformations,...), ciliary deficiency and WDR11/cilia colocalization. Using in vitro experiments, they show the formation of a tertiary complex composed of Gli3-WDR11-EMX and claim WDR11 to enhance Hh-induced expression of GnRH. Importantly, they report a new WDR11 mutation in human leading to defective nuclear localization and EMX binding causing both pituitary dysgenesis and obesity. Finally they show preliminary in vitro data suggesting the use of Hh agonists to rescue WDR11 deficiency.

While we find the paper contains extensive characterization of the WDR11 KO/KD with direct clinical relevance to human disease, we feel that the manuscript cannot be published in the present form as a solidified mechanistic link between WDR11 and Hh pathway remains lacking. For clarity, we would suggest the authors focus on the role of WDR11 in Gli3/EMX complex in chromatin regulation/Hh target gene expression (through WDR11 ChIP experiment for example). Also Immunostaining quality needs to be improved and interpretation of some of the figures could be made easier.

Major comments:

1. The title should be changed as WDR11 was not shown to directly disrupt Hh signaling in the manuscript.

Response: We agree with the reviewer about the potential overstatement. We have now changed the title to “WDR11-mediated Hedgehog signalling defects underlie a new ciliopathy related to Kallmann syndrome.” In addition, we also changed the title of Figure 6 as “WDR11 cooperates with Hh pathway to regulate gene expression”.

2. WDR11 KO validation at the protein level is performed on kidney tissue (Fig. 1A). As the paper's main focus is on ME and OB, we would suggest to validate WDR11 KO in brain tissue.

Response: The absence of WDR11 protein in ME of the KO mouse brain was already shown in Appendix Figure S 2C. We now include additional OB tissue staining data (Expanded View Figure EV1A) and western blot data using lysates from the testis and brain (Fig. 1A).

3. Overall there is inconsistency in cilia staining; In general, it is best to use both axonemal and basal body co-stainings when one counts cilia. For key quantitative figures, please include at least one representative image with both stained.

Response: We initially used co-staining (double marker) to count cilia in MEFs (Fig 3A) and zebrafish morphants (Appendix Figure S5F). Both of these data demonstrated that Wdr11 KO/KD caused the disruption of axoneme, not the basal body. Since WDR11 did not affect the formation of basal body, we then focused on the status of axoneme for the rest of our study. First, we determined the localisation of WDR11 in relation to the ciliary structures in vivo and in vitro. Here, we used single staining for either the axoneme (acetylated tubulin) or the basal bodies/centrosomes (g-tubulin) in order to differentially visualise WDR11 with an alternative fluorescein labelling. This study revealed prominent staining of WDR11 around the basal bodies, with occasional axonemal distribution (Fig. 3C, Fig. 4I). Next, we moved on to analyse the effects of Wdr11 KO regarding the frequency, length and structure of axoneme. Acetylated tubulin staining was mostly used here to demonstrate the defective axoneme formation, which was the primary defect in the KO mouse brain and zebrafish morphants (Fig. 3B, Fig. 4G and H, and Expanded View Figure EV1 A and C). To clarify, we have now reorganised Figure 3 and included additional representative images co-stained for markers of axoneme (Arl13b) and basal body (g-tubulin) to further substantiate the data. We also include the representative cell images (Appendix Fig S9) used to generate the graphs in Figure 8C. We also modified the text (page 8 & 10).

4. The authors show WDR11-dependency of Hh pathway target gene expression (ptch1 and Gli3); however WDR11 cytoplasmic localization upon Hh pathway activation (fig 3D and suppl fig 5A) does not support the functional role of WDR11 binding to nuclear EMX (Fig 5). Rather, it supports

the opposite-that WDR11 inhibits EMX binding to Gli3. Please clarify.

Response: WDR11 is a chaperone protein that traffics from the cilia through the cytoplasm to the nucleus, likely mediated by its interaction with GLI3. There is a constant “nuclear influx” of WDR11, as evidenced by the distinct nuclear accumulation of WDR11 when the nuclear export is blocked (Leptomycin treatment). Furthermore, the disease-associated WDR11 mutants showed a defective nuclear influx capacity (Fig. 7D and Appendix Fig S7C). As shown in Figure 3C and D, WDR11 is localised in the cytoplasm in the majority (~65%) of the cases, but the nuclear WDR11 always exists even under basal (unstimulated) conditions. This ratio of cytoplasmic and nuclear distribution does not change upon Hh signal activation (purmorphamine treatment). However, we believe it is the “status of GLI3” bound to WDR11 that matters. It has been previously established that the *active* GLI proteins are unstable and only transiently stay in the nucleus, showing a similar staining pattern as WDR11 in our current study. The *inactive* form of GLI (GLI R), as shown with cyclopamine treatment, tends to be more stable and accumulates in the nucleus (Fig 3C and D). The nuclear accumulation of *inactive* GLI is believed to be important in maintaining the “OFF state” of the pathway in the absence of Hh ligand. Our data suggest that WDR11 does not sequester GLI3 in any specific cellular compartment, but rather stabilises GLI3 as a scaffolding partner. Overexpression of WDR11 increases both GLIFL and GLIR protein levels (Fig 5D). We hypothesise that the movement of WDR11 is dependent on the status of the GLI3, which is transient and responsive to Hh signalling. WDR11 does not behave like Suppressor-of-Fused (Sufu) which has an antagonistic role in Hh signalling by sequestering GLI proteins in the cytoplasm and increasing nuclear export. We hypothesise that WDR11 enhances Hh signalling by supporting the interaction of GLI with other proteins such as EMX1.

To clarify, we now include additional images representing the various cellular localisation and trafficking behaviours of WDR11 (Appendix Fig S6A). We have also modified the text describing the effects of human mutations on WDR11 nuclear accumulation (page 14).

5. The authors claim that WDR -/- mice present obesity, a hallmark of BBS-type ciliopathies.

However, in the actual data (suppl. fig 3A), the data shows they have lower weight, a seeming contradiction that weakens the authors ciliopathy claim. This contradiction needs to be resolved.

Response: The growth curve we initially presented in Appendix Figure S4A was based only on body weight. However, when we compared their body mass index (= weight/length²), because the mutant mice were shorter than the WT, they exhibited statistically higher BMI. This was in line with the clinical cases where the two boys with MT mutation were shorter than normal, resulting in a high BMI score (Appendix Text and Table S3).

To clarify, we have now modified the graphs (Fig. 2G) and text (page 7). We also include the actual growth parameters of these mice showing their weight and length (Appendix Table S3). We also insert the photos of the mice in Appendix Figure S4C to illustrate the short stature of these mice.

Minor comments:

1. For clarity, we would suggest to report and discuss anatomic considerations regarding pituitary gland (shown in Fig 2F) and pituitary functionality (Fig 2D) before potential consequences (delayed puberty, reproductive dysfunctions, obesity) (discussed in Fig.2A-B-C-G).

Response: We agree with the Reviewer and changed the order (page 6 - 7).

2. Pituitary gland functionality (GH, PRL, LH, FSH) was done at the mRNA level in the WDR11 KO mice (Fig.2D); we would suggest to confirm these results at the protein level, given previous demonstration of post transcriptional regulation.

Response: All of our KS/CHH patients with WDR11 mutations (M1-M5) showed low plasma levels of LH and FSH upon diagnosis, as reported in our previous publication (Kim *et al. Am J Hum Genet.* 87(4):465-79. doi: 10.1016/j.ajhg.2010.08.0180). The index patient with WDR11 mutation MT showed a clinical GH deficiency with low serum levels of IGF/IGFBP (Appendix Text). Wdr11 KO mice exhibit the classical phenotypes of pituitary hormone deficiency. Combined, these data support the notion that these hormones are functionally reduced at protein levels in vivo, and the possible “rescue” by post-transcriptional regulation seems quite unlikely. However, we do agree with the Reviewer in that we did not provide any evidence for the functional expression of PRL. We now include the PRL immunofluorescence data of mouse brain sections, demonstrating the significant reduction of PRL protein in the Wdr11^{-/-} pituitary glands (Appendix Figure S3B).

3. *Although the authors downplay the result, the finding of reduced GnRH neurons in WDR11 KO brains is an important finding! Please provide an illustrating figure (fig 2E) and include a discussion of its significance.*

Response: We appreciate the Reviewer's encouraging comments on the importance of this finding. The total number of GnRH neurons were determined by counting the positive immunoreactivity in serially sectioned whole embryo head. We now include the representative images from these studies (Appendix Figure S2A). We also include further data suggesting that WDR11 KO may cause delayed migration of GnRH neurons. We observed a relative increase of GnRH neurons under the cribriform plate area versus forebrain regions in the *Wdr11*^{-/-} brain, compared to WT (Appendix Figure S3A). This is now discussed in the main text (page 6).

4. *The authors claim fatty livers in WDR +/- mice, although liver histology is not shown*

Response: The histology data of the Oil Red O-stained mouse liver sections, demonstrating the accumulation of vacuolated microvesicular fat are now included (Appendix Figure S4D).

5. *In Fig. 3, the authors claim the colocalization of WDR11 and cilia in OB and ME ; cilia deficiency is however reported in the choroid plexus ; for consistency, we suggest the authors show cilia deficiency in OB and ME.*

Response: We now include cilia staining data of OB and ME (Expanded View Figure EV.1A and C). We reported the choroid plexus because the null mice showed hydrocephalus phenotypes, known to be caused by the defective cilia in this part of the brain. We believe this adds the clinical implication of our study as ciliopathy affects many organs, not just the HPG axis.

6. *In Fig. 3a WDR11 staining looks cytoplasmic whereas it is nuclear in Suppl. Fig. 2a ; please comment on this surprising difference if representative.*

Response: As mentioned above, WDR11 is a chaperoning and scaffolding protein that traffics from the cilia, cytoplasm and nucleus. Therefore, its localisation in tissues could also be variable, depending on the status of the cell cycle (as ciliogenesis is inhibited during mitosis) or Hh signal activity. Figure 3 aimed to demonstrate WDR11 localisation in relevance to the cilia, thus the cells showing cytoplasmic/cilia staining were mainly shown. Appendix Figure S2 aimed to determine the co-localisation of WDR11 with GnRH staining. As previously reported by other investigators in the field, GnRH staining often appears stronger in the nucleus with a diffuse cytoplasmic signal. We now include more representative images from the nasal cavity area to further substantiate our conclusion (Appendix Fig S2).

7. *Suppl. fig 2b would benefit from a higher magnification picture ; why did the authors use WDR +/- instead of WDR11 +/- ? Is there a difference in heterozygotes versus wildtype?*

Response: We agree with the Reviewer and now include the higher magnification images in Appendix Fig S2B. We apologise for the confusion caused by typing errors - the top row of the top panel was mistakenly labelled as +/-, instead of +/+; the top row of the bottom panel was mistakenly labelled as +/+, instead of +/- . Heterozygotes rarely display minor anatomical differences compared to the WT. To clarify, we have now removed the heterozygotes and x-GAL staining images in this figure.

8. *In vitro WDR11/cilia colocalization is shown using exogenous WDR11 expression in 3T3. It would be more appropriate to use cell lines spontaneously expressing WDR11 (as GnRH neuronal cell lines)*

Response: We have demonstrated ciliary localisation of endogenous WDR11 in vivo using mouse and zebrafish tissue sections (Expanded View Figure EV1, Fig 3 and Fig 4). The endogenous level of WDR11 protein in GnRH neuronal cell lines in vitro are rather low. Moreover, the dynamic and broad distribution of WDR11 as discussed above, makes it practically impossible to rely on the endogenous protein for cilia localisation studies. We believe our current data from animal tissues and in vitro overexpression studies provide complementary and sufficient evidence to support the notion that WDR11 associates with cilia.

9. *For consistency with the murine WDR11 KO, we would suggest to test for WDR11 expression in zebrafish OB, ME and pronephros, as well as potential colocalization with cilia (fig 4G-H)*

Response: We now include new data demonstrating *wdr11* expression and colocalisation with cilia in the zebrafish OB (Fig 4I). In situ hybridisation data of *wdr11* in different stages of zebrafish

development was reported in our previous publication (Kim *et al. Am J Hum Genet.* 87(4):465-79. doi: 10.1016/j.ajhg.2010.08.0180), which indicated ubiquitous expression at 24hpf.

10. Fig 4E-F-G-H show morpholino E3I3 phenotype; for consistency we would suggest to show E9I9 phenotype too (supplemental data)

Response: We now include new data (Appendix Figure S5 C, D and F) and discuss them in the text (page 9).

11. GnRH expression upon Hh pathway activation in GnRH neuronal cell lines should be confirmed at the protein level (Fig. 6)

Response: We now include the GnRH immunofluorescence data of GN11 cells with or without purmorphamine treatment, confirming the induction of GnRH protein expression upon Hh signal activation, which is consistent with our qRT-PCR data. We also include the GnRH immunofluorescence data in GN11 cells after WDR11 transfection, confirming that overexpression of WDR11 is sufficient to increase GnRH protein levels (Appendix Figure S8).

12. The authors show that WDR 11 is sufficient to enhance Hh pathway-induced GnRH expression in the presence of Gli3/EMX ; WDR11 necessity for Hh pathway-induced GnRH expression in the presence of Gli3/EMX should be tested (using either WDR-/- MEF or WDR11KD in GnRH neuronal cell lines)

Response: We have already used FNCB4-hTERT cells with shRNA-mediated knockdown of WDR11 and showed that WDR11 is required for Hh-dependent induction of GnRH and other Hh pathway genes. We have also shown the defective expression and induction of Hh pathway genes in Wdr11 null MEFs (Fig 6). Please note MEFs do not express GnRH.

To further substantiate, we now include additional data for GLI3 and EMX1 induction in WDR11-knockdown FNCB4-hTERT cells (Fig 6B) and discussed this in the text (page 11).

13. Both the mother and 2 brothers are heterozygous for the WDR11 described mutation; the mother however does not present any phenotype. Please comment on disease penetrance.

Response: The penetrance of the p.Pro537Leu mutation seems to be reduced, because the mother, who also carries the mutation, had normal brain MRI. We have now elaborated the definition of the phenotype, which is based on the MRI findings, and discussed the incomplete penetrance of the mutation in the main text (page 13). It is notable that variable penetrance is seen even in Wdr11 homozygous null mice.

14. As previously mentioned, GnRH induction upon Hh pathway activation should be confirmed at the protein level in Fig.8A. Please indicate what are the cells used in the experiment.

Response: We now include the GnRH immunofluorescence staining data in GN11 cells after purmorphamine treatment or WDR11 transfection, confirming the induction of GnRH at protein level (Appendix Fig. S8). The name of the cell line was stated in the main text, but not in the figure legend. We are sorry about this error, which is now corrected.

15. The authors claim WDR11 expression in the « GnRH migratory niche including nasal cavity and cribriform plate », this is not shown in fig. 1B. The authors should either show the data or remove the reference in the paper.

Response: WDR11 expression within the nasal cavity and cribriform plate were presented in Appendix Fig S2 but may not have been explained enough. We now include additional images demonstrating the expression of WDR11 in the nose area and the forebrain regions with clear labelling of the anatomical structures (Appendix Fig S1C, S2, S3A) to support our statement.

16. Fig 3A low quality pictures

Response: We now include new images (Expanded View Figure EV1).

17. Fig 3C would benefit from a lower magnification (more than 1 cell).

Response: We now include additional images at lower magnification showing multiple cells (Fig 3A).

18. Fig 5D: determined instead of « determiend »

Response: The typo is corrected.

19. Scale bars are missing in various figures, or figure legends

Response: Scale bars are now included in all figures and figure legends.

20. Error bars represent either SEM or SD ; this should be uniformed in the whole manuscript

Response: We now use SEM throughout the manuscript.

21. Suppl Fig. 3A, WDR11 +/- are in the legend, but not in the figure

Response: We are sorry for this error. It is now removed from the legend.

Referee #2:

1. This is an interesting paper with solid data on the mouse KO and fish morphants. The links with HPE, cilia function in Hh signaling and Gli3 isoform formation are intriguing and worth publishing in EMBO Reports.

Response: We appreciate the Reviewer's positive comments.

2. The last needs cleaner Westerns and full size blots, instead of cutting the windows of interest (e.g. Fig. 7A).

Response: The original scans of the entire blots/gels are now submitted as Source Data Files.

3. The presentation of the can be improved. The text, figures etc need revising. Citations of the figures in the text appear not to be always consistent. The figures and/or the text must be organized better. Figures do not appear in order along the text. For instance, Figure 3A is named first than figure 2. Figure 4 is before than Figure 3D-E.

Response: We have now revised and reorganised the text and figures to make the transitions smoother.

4. Unlike what is stated in the text, Figure 1B does not show a detailed analysis of the brain.

Response: Further analysis of the nasal cavity and cribriform plate were shown in Appendix Fig S2, not in Fig 1, but may not have been explained enough. We now include additional images of WDR11 expression in the nose and the forebrain regions with clear labelling of the anatomical structures to support our statement (Appendix Fig S1C, S2, S3A).

5. Figure 3A. I do not see the correspondence between the full image and the zoomed region.

Response: We now include new images to show the correlation better (Expanded View Figure EV. 1C).

Referee #3:

Summary:

1. Does this manuscript report a single key finding?

YES. It reports that, in vertebrate development, the scaffold protein WDR11 is required for ciliogenesis through the Hedgehog signaling pathway, based on knock-out mice and zebrafishes and cellular and molecular analysis.

2. Is the reported work of significance?

YES, since the generation and the analysis of knock-out animals are original. But the importance of WDR11 in development of mice and human has already been documented using other methods, such as the discovery of human genetic mutations.

3. Is it of general interest to the molecular biology community?

YES. It is important to elucidate the function of the protein WDR11.

4. Is the single major finding robustly documented using independent lines of experimental evidence?

YES. The analysis of the knock-out animals is well-performed although the number of animals

analyzed seems a little weak (5 or below in each group, when indicated).

The manuscript by Kim et al. analyzes the functional role of the protein WDR11, a WD protein that interacts with transcription factor EMX1. Recently, this protein was found to be mutated in congenital hypogonadotropic hypogonadism (CHH) such as Kallmann syndrome (KS). This protein contains twelve WD domains that form two beta propellers potentially serving as molecular scaffolds. But the role of WDR11 in development remains unclear. Here, the authors report that WDR11 modulates the Hedgehog (Hh) signaling pathway and is essential for ciliogenesis. They have generated both knockout (KO) mice and zebrafishes to investigate the function of WDR11 in vertebrate development and ciliogenesis. After a deep anatomical characterization of the Wdr11 KO mice and KO fishes and a cellular analysis of the WDR11, they have concluded for an important role of Wdr11 in Hh signaling and ciliogenesis. Then by co-immunoprecipitation experiments from transfected HEK-293 and Wdr11 KO MEF cells, they have brought evidence that WDR11 forms a tertiary complex with EMX1 and GLI3 transcription factors. The WDR11 was essential for EMX1-GLI3 interaction. In GnRH neuronal cell lines, the authors propose that WDR11 mediates the expression of these two transcription factors as well as GNRH1 through the activation of the Hh signaling pathways by the agonist purmorphamine. In these cell lines, they also show that co-expression of Gli1 and Gli3 proteins, but not Gli2, induced EMX1 expression (but not of WDR11) indicating that the transcription factors GLI target EMX1 but not WDR11. In addition, they have identified one Kallmann patient with a new genetic mutation in WDR11 (heterozygous). They have analyzed the functional consequence of this mutation in WDR11 together with other genetic mutations previously reported in this protein. All these mutants impaired WDR11 function, such as its translocation to the nucleus and the formation of the ternary complex with EMX1 and GLI3. Finally, they have tested the effect of purmorphamine in rescuing the expression of GNRH1 in GnRH neuronal cell lines treated with WDR11 shRNA.

Many results are of interest and novel. The manuscript reports that, in vertebrate development, the scaffold protein WDR11 is required for ciliogenesis through the Hedgehog signaling pathway, based on knock-out mice and zebrafishes and cellular and molecular analysis. The generation and the analysis of knock-out animals is original. But the importance of WDR11 in development of mice and human has already been documented using other methods. The results are appropriately discussed in the context of earlier literature. They will be of interest for investigators working in field of hypothalamus-pituitary development and more generally in the biology of cilia. WDR11 seems to be a key scaffold protein in these processes, and knock-out animal is a key tool to understand the function of this protein. In general, the experimental data are of sufficient quality to justify the conclusions, and the manuscript is clearly written.

Response: We appreciate the positive comments from the Reviewer.

Specific comments:

1. The abstract is confusing and some part should be deleted and rephrased. They should remove the part "that are affected not only in CHH/KS...leads to obesity both in mice and men". Indeed, it's confusing to mix the results of the study with interpretation of animal and human phenotype. Also, the authors should remove the text "as one of key links between reproduction and metabolism" and conclude more about their results.

Response: We now include the revised abstract.

2. The authors do not clearly state how many animals (both mice and fishes) were used in the experiments. For example, they used a small number (3 to 5) of Wdr11-null males (see page 6). It would be better to use in most cases at least 5 animals. It should be clarified.

Response: We now state the exact numbers of animals in all relevant figure legends. We also include data obtained from additional animals, where the numbers were less than 5 (Fig 2 C and G, Appendix Fig. S4).

3. Figure 2, panel D: could you explain the large differences between males and females for GnRHR, LH and FSH expression in the KO animals?

Response: We thank the Reviewer for bringing up this interesting point. We think the data could suggest a potential sexual dimorphism - females are more severely affected by WDR11 than the

males, at least in the pituitary phenotypes. We now include a new sentence in the text to discuss this (page 7).

4. *Co-immunoprecipitation was done with recombinant WDR11, EMX1 and GLI3 after cell transfections. However, primary antibodies against these proteins have been used (see Materials and Methods). So why the authors did not show co-immunoprecipitation experiments with the endogenous proteins? Would it be possible to do some of these experiments, since such data will reinforce the existence of the ternary complex proposed in native conditions?*

Response: We now include the additional co-IP western blot data obtained from mouse tissues, demonstrating the binding interactions of endogenous proteins (Fig5. B).

5. *Title Figure 8: remove "therapeutic" since it is not adapted here.*

Response: The word is now omitted from the title of Fig 8.

6. *How selective the drug purmorphamine is in term of drug target due to its low potency (micromolar, see Fig. 8A)? It would be an information useful for the readers outside of the field of Hh signaling. Do better agonists are existing?*

Response: Purmorphamine is a tri-substituted purine derivative that directly binds to Smoothed with an $EC_{50} = 1 \mu\text{M}$ (see reference: *Sinha S & Chen JK (2006) Nat Chem Biol. 2:29-30*). SAG (Smoothed Agonist) is a chlorobenzothiophene-containing compound which activates Smoothed by binding to the heptahelical bundle ($K_d = 59 \text{ nM}$) and stabilizing a specific conformation of SMO with an $EC_{50} = 3 \text{ nM}$ (reference: *Chen JK et al. (2002) PNAS 99:14071-14076*). We now include a short paragraph discussing this in the main text (page 17).

2nd Editorial Decision

8 November 2017

Thank you for the submission of your revised manuscript to EMBO reports. I contacted referee 1 and 3 to review the revised version of your manuscript. Unfortunately, referee 1 was not available so I asked referee 3 to assess the manuscript also taking the other referee's concerns into account. We have now received the report from former referee 3, which is copied below.

As you will see this referee is now positive about the study and supports its publication in EMBO reports.

REFEREE REPORT

Referee #3:

The authors have clearly addressed the concerns pointed out by the different referees. They have largely reorganized the figures and modified the main text, and they have included some additional data. The manuscript is acceptable for publication.

YOU MUST COMPLETE ALL CELLS WITH A PINK BACKGROUND

Corresponding Author Name: Soo Hyun Kim
 Journal Submitted to: EMBO Reports
 Manuscript Number: 2017-44632-T